# Iteratively Reweighted Least Squares for Basis Pursuit with Global Linear Convergence Rate

**Christian Kümmerle**[*§]
Department of Applied Mathematics & Statistics
Johns Hopkins University

**Claudio Mayrink Verdun**[†§]
Department of Mathematics and Department of Electrical and Computer Engineering
Technical University of Munich

**Dominik Stöger**[‡§]
Department of Mathematics
KU Eichstätt–Ingolstadt

## Abstract

The recovery of sparse data is at the core of many applications in machine learning and signal processing. While such problems can be tackled using $\ell_1$-regularization as in the LASSO estimator and in the Basis Pursuit approach, specialized algorithms are typically required to solve the corresponding high-dimensional non-smooth optimization for large instances. Iteratively Reweighted Least Squares (IRLS) is a widely used algorithm for this purpose due to its excellent numerical performance. However, while existing theory is able to guarantee convergence of this algorithm to the minimizer, it does not provide a global convergence rate. In this paper, we prove that a variant of IRLS converges *with a global linear rate* to a sparse solution, i.e., with a linear error decrease occurring immediately from any initialization, if the measurements fulfill the usual null space property assumption. We support our theory by numerical experiments showing that our linear rate captures the correct dimension dependence. We anticipate that our theoretical findings will lead to new insights for many other use cases of the IRLS algorithm, such as in low-rank matrix recovery.

## 1 Introduction

The field of sparse recovery deals with the problem of recovering an (approximately) sparse vector $x$ from only few linear measurements, presented by an underdetermined system of linear equations of the form $y = Ax$. One approach to solve this problem is to consider the $\ell_0$-minimization under linear constraints, which is NP-hard in general [24, 54]. For computational reasons, instead of $\ell_0$-minimization, it is common practice to consider its convex relaxation

$$\min_{x \in \mathbb{R}^N} ||x||_1 \qquad \text{subject to } Ax = y, \qquad (P_1)$$

---

[*]kuemmerle@jhu.edu

[†]verdun@ma.tum.de

[‡]dominik.stoeger@ku.de

[§]The name order of the authors is alphabetical.

35th Conference on Neural Information Processing Systems (NeurIPS 2021).

where $A \in \mathbb{R}^{m \times N}$, $b \in \mathbb{R}^m$ are given, which is referred to as $\ell_1$(*-norm) minimization* [20, 27, 64] or *basis pursuit* [18, 19] in the literature.

Unlike $\ell_0$-minimization, the optimization program $(P_1)$ is computationally tractable in general, and a close relationship of their minimizers has been recognized and well-studied in the theory of compressive sensing [12, 13, 26, 32]. In statistics and machine learning, an unconstrained variant of $(P_1)$, often called *LASSO*, amounts to the most well-studied tractable estimator for variable selection in high-dimensional inference [37, 48, 65]. $\ell_1$-minimization has many other applications and it was even called the *modern least squares* [14].

The tractability of $(P_1)$ becomes evident from the fact that it can be reformulated as a linear program [66]. However, as many problems of interest in applications are high-dimensional and therefore challenging for standard linear programming methods, many specialized solvers for $(P_1)$ have been proposed, such as the Homotopy Method [28], primal-dual methods [15, 52], Alternating Direction Method of Multipliers [9], Bregman iterative regularization [71] and Semismooth Newton Augmented Lagragian Methods [46] and Iteratively Reweighted Least Squares (IRLS), the latter of which is in the focus of this paper.

Iteratively Reweighted Least Squares corresponds to a family of algorithms that minimizes non-smooth objective functions by solving a sequence of quadratic problems, with its idea going back to a method proposed by Weiszfeld for the *Fermat-Weber problem* [7, 69]. A variety of different problems such as robust regression in statistics [39, 53], total variation regularization in image processing [2, 33, 55], joint learning of neural networks [72], robust subspace recovery [45] and the recovery of low-rank matrices [31, 42, 43, 51] can be solved efficiently by IRLS in practice, as it relies on simple linear algebra to solve the linear systems arising from the quadratic problems at each iteration, without the need of a careful initialization or intricate parameter tuning. On the other hand, the analysis of IRLS methods is typically challenging: General convergence results are often weak, and stronger convergence results are only available in particular cases; see Section 2.3 for more details.

**IRLS for sparse recovery.** In the sparse recovery context, the first variants of IRLS were introduced in [34, 59] for the $\ell_p$-quasinorm minimization problem $(P_p)$ with $0 < p \leq 1$ that is similar to $(P_1)$, but with $\|x\|_p$ instead of $\|x\|_1$ as an objective. In [16], modifications of the method of [34, 59] using specific smoothing parameter update rules were observed to exhibit excellent numerical performance for solving $(P_p)$, retrieving the underlying sparse vector when most of the methods fail. A useful fact is that IRLS is one of the few methods (ADMM being the other one [9]) that provides a framework to solve both constrained and unconstrained formulations of $\ell_p$-minimization problems.

A major step forward in the theoretical understanding of IRLS was achieved in [23], where the authors showed that a variant of IRLS for $(P_1)$ converges globally to an $\ell_1$-minimizer if the measurement operator $A$ fulfills the null space property of sufficient order, which essentially ensures that an $\ell_1$-minimizer is actually sparse. However, since this proof relies on the existence of a convergent subsequence, their proof does not reveal any rate for *global* convergence. The analysis of [23] provides, furthermore, a *locally* linear convergence rate, but this local linear rate has the drawback that it only applies *if the support of the true signal has been discovered*, which is arguably the difficult part of $\ell_0$-minimization—cf. Proposition 3.1 below and Section 4.1.

While several extensions and modifications of the IRLS algorithm in [23] have been proposed (see, e.g., [3, 30]), this following fundamental algorithmic question has remained unanswered:

> *What is the global convergence rate of the IRLS algorithm for $\ell_1$-minimization?*

**Our contribution.** We resolve this question, formally stated in [62], and present a new IRLS algorithm that *converges linearly* to a sparse ground truth, *starting from any initialization*, as stated in Theorem 3.2. Our algorithm returns a feasible solution with $\delta$-accuracy, i.e., $\|x_* - x^k\|_1 \leq \delta$, where $x_*$ is the underlying $s$-sparse vector, in $k = O(N\sqrt{(\log N)/m} \log(1/\delta))$ iterations. Analogous to [23], it is assumed that the measurement matrix $A$ satisfies the so-called null space property [21]. We also provide a similar result for approximately sparse vectors. Our proof relies on a novel quantification of the descent of a carefully chosen objective function in the direction of the ground truth. Additionally, we support the theoretical claims by numerical simulations indicating that we capture the correct dimension dependence. We believe that the new analysis techniques in this paper are of independent interest and will pave the way for establishing global convergence rates for other variants of IRLS such as in low-rank matrix recovery [31].

**Notation.** We denote the cardinality of a set $I$ by $|I|$ and the support of a vector $x \in \mathbb{R}^N$, i.e., the index set of its nonzero entries, by $\text{supp}(x) = \{j \in [N] : x_j \neq 0\}$. We call a vector $s$-sparse if at most $s$ of its entries are nonzero. We denote by $x_I$ the restriction of $x$ onto the coordinates indexed by $I$, and use the notation $I^c := [N] \setminus I$ to denote the complement of a set $I$. Furthermore, $\sigma_s(x)_{\ell_1}$ denotes the $\ell_1$-*error of the best $s$-term approximation* of a vector $x \in \mathbb{R}^N$, i.e., $\sigma_s(x)_{\ell_1} = \inf\{\|x - z\|_1 : z \in \mathbb{R}^N \text{ is } s\text{-sparse}\}$.

## 2   IRLS for sparse recovery

We now present a simple derivation of the Iteratively Reweighted Least Squares (IRLS) algorithm for $\ell_1$-minimization which is studied in this paper. IRLS algorithms can be interpreted as a variant of a Majorize-Minimize (MM) algorithm [63], as we will lay out in the following. It mitigates the non-smoothness of the $\|\cdot\|_1$-norm by using the smoothed objective function $\mathcal{J}_\varepsilon : \mathbb{R}^N \to \mathbb{R}$, which is defined, for a given $\varepsilon > 0$, by

$$\mathcal{J}_\varepsilon(x) := \sum_{i=1}^N j_\varepsilon(x_i) \quad \text{with} \quad j_\varepsilon(x) := \begin{cases} |x|, & \text{if } |x| > \varepsilon, \\ \frac{1}{2}\left(\frac{x^2}{\varepsilon} + \varepsilon\right), & \text{if } |x| \leq \varepsilon. \end{cases} \quad (1)$$

The function $\mathcal{J}_\varepsilon$ can be considered as a scaled Huber loss function which is widely used in robust regression analysis [40, 50]. Moreover, the function $\mathcal{J}_\varepsilon$ is continuously differentiable and fulfills $|x| \leq j_\varepsilon(x) \leq |x| + \varepsilon$ for each $x \in \mathbb{R}$. Instead of minimizing the function $\mathcal{J}_\varepsilon$ directly, the idea of IRLS is to minimize instead a suitable chosen quadratic function $Q_\varepsilon(\cdot, x)$, which majorizes $\mathcal{J}_\varepsilon$ such that $Q_\varepsilon(z, x) \geq \mathcal{J}_\varepsilon(z)$ for all $z \in \mathbb{R}^N$. This function is furthermore chosen such that $Q_\varepsilon(x, x) = \mathcal{J}_\varepsilon(x)$ holds, which implies that $\min_{z \in \mathbb{R}^n} Q_\varepsilon(z, x) \leq \mathcal{J}_\varepsilon(x)$. The latter inequality implies that by minimizing $Q_\varepsilon(\cdot, x)$, IRLS actually achieves an improvement in the value of $\mathcal{J}_\varepsilon$ as well. More specifically, $Q_\varepsilon(\cdot, x)$ is defined by

$$\begin{aligned} Q_\varepsilon(z, x) &:= \mathcal{J}_\varepsilon(x) + \langle \nabla\mathcal{J}_\varepsilon(x), z - x\rangle + \frac{1}{2}\langle(z - x), \text{diag}(w_\varepsilon(x))(z - x)\rangle \\ &= \mathcal{J}_\varepsilon(x) + \frac{1}{2}\langle z, \text{diag}(w_\varepsilon(x))z\rangle - \frac{1}{2}\langle x, \text{diag}(w_\varepsilon(x))x\rangle, \end{aligned} \quad (2)$$

where $\nabla\mathcal{J}_\varepsilon(x) = \left(\begin{cases} \frac{x_i}{|x_i|}, & \text{if } |x_i| > \varepsilon \\ \frac{x_i}{\varepsilon}, & \text{if } |x_i| \leq \varepsilon \end{cases}\right)_{i=1}^N$ is the gradient of $\mathcal{J}_\varepsilon$ at $x$ and the weight vector $w_\varepsilon(x) \in \mathbb{R}^N$ is a vector of *weights* such that $w_\varepsilon(x)_i := [\max(|x_i|, \varepsilon)]^{-1}$ for $i \in [N]$. The following lemma shows that $Q_\varepsilon(\cdot, \cdot)$ has indeed the above-mentioned properties. We refer to the supplementary material for a proof.

**Lemma 2.1.** *Let $\varepsilon > 0$, let $\mathcal{J}_\varepsilon : \mathbb{R}^N \to \mathbb{R}$ be defined as in (1) and $Q_\varepsilon : \mathbb{R}^N \times \mathbb{R}^N \to \mathbb{R}$ as defined in (2). Then, for any $z, x \in \mathbb{R}^N$, the following affirmations hold:*

*i.* $\text{diag}(w_\varepsilon(x))x = \nabla\mathcal{J}_\varepsilon(x)$,    *ii.* $Q_\varepsilon(x, x) = \mathcal{J}_\varepsilon(x)$,        *iii.* $Q_\varepsilon(z, x) \geq \mathcal{J}_\varepsilon(z)$.

As can be seen from the equality in (2), minimizing $Q_\varepsilon(\cdot, x)$ corresponds to a minimization of a *(re-)weighted least squares objective* $\langle \cdot, \text{diag}(w_\varepsilon(x))\cdot\rangle$, which lends its name to the method. Note that unlike a classical MM approach, however, IRLS comes with an *update* step *of the smoothing parameter* $\varepsilon$ at each iteration. We provide an outline of the method in Algorithm 1.

The weighted least squares update (3) can be computed such that $x^{k+1} = W_k^{-1}A^*(AW_k^{-1}A^*)^{-1}(y)$ with $W_k = \text{diag}(w_k)$, with the solution of the $(m \times m)$ linear system $(AW_k^{-1}A^*)z = y$ as a main computational step. This linear system is positive definite and suitable for the use of iterative solvers. In [30], an analysis of how accurately the linear system of a similar IRLS method needs to be solved to ensure overall convergence. We note that for small $\varepsilon_k$, the Sherman-Woodbury formula [70] can be used so that the calculation of $x^{k+1}$ boils down to solving a smaller linear system that is well-conditioned, c.f. the supplementary material for details. This numerically advantageous property is not shared by the methods of [3, 23, 30], as our smoothing update (4) is slightly different from the ones proposed in these papers. We refer to Section 2.2 for a discussion.

The update step of the smoothing parameter $\varepsilon$ (4) for the IRLS algorithm under consideration requires an a priori estimate of the sparsity of the ground truth of the signal, a piece of information that is

**Algorithm 1** Iteratively Reweighted Least Squares for $\ell_1$-minimization

---

**Input:** Measurement matrix $A \in \mathbb{R}^{m \times N}$, data vector $y \in \mathbb{R}^m$,
initial weight vector $w_0 \in \mathbb{R}^N$ (default: $w_0 = (1, 1, \ldots, 1)$).
Set $\varepsilon_0 = \infty$.
**for** $k = 0, 1, 2, \ldots$ **do**

$$x^{k+1} := \underset{z \in \mathbb{R}^N}{\arg\min} \langle z, \mathrm{diag}\,(w_k)\, z \rangle \quad \text{subject to} \quad Az = y, \tag{3}$$

$$\varepsilon_{k+1} := \min \left( \varepsilon_k, \frac{\sigma_s(x^{k+1})_{\ell_1}}{N} \right), \tag{4}$$

$$(w_{k+1})_i := \frac{1}{\max \left( |x_i^{k+1}|, \varepsilon_{k+1} \right)} \qquad \text{for each } i \in [N], \tag{5}$$

**end for**
**return** Sequence $(x^k)_{k \geq 1}$.

---

also needed by most of the methods for sparse reconstruction. In practice, an overestimation of $s$ is not a problem for similar numerical results if the overestimation remains within small multiples of the sparsity of the signal. We note, however, that there are also versions of IRLS which do not require a-priori knowledge of $s$, e.g. [30, 68], as the update rule for the smoothing parameter is chosen differently. An interesting future research direction is to extend the analysis presented here to IRLS with such a smoothing parameter update.

A consequence of Lemma 2.1, step (4), the fact that $\varepsilon \mapsto \mathcal{J}_\varepsilon(z)$ is monotonously non-decreasing, and that $k \mapsto \varepsilon_k$ is non-increasing is that $k \mapsto \mathcal{J}_\varepsilon(z)$ is non-increasing in k. This implies that the iterates $x^k, x^{k+1}$ of Algorithm 1 fulfill

$$\mathcal{J}_{\varepsilon_{k+1}}(x^{k+1}) \leq \mathcal{J}_{\varepsilon_k}(x^{k+1}) \leq Q_{\varepsilon_k}(x^{k+1}, x^k) \leq Q_{\varepsilon_k}(x^k, x^k) = \mathcal{J}_{\varepsilon_k}(x^k). \tag{6}$$

This shows in particular that the sequence $\left\{ \mathcal{J}_{\varepsilon_k}\left( x^k \right) \right\}_{k=0}^{\infty}$ is non-increasing. For this reason, it can be shown that each accumulation point of the sequence of iterates $(x^k)_{k \geq 0}$ is a (first-order) stationary point of the smoothed $\ell_1$-objective $J_{\overline{\varepsilon}}(\cdot)$ subject to the measurement constraint imposed by $A$ and $y$, where $\overline{\varepsilon} = \lim_{k \to \infty} \varepsilon_k$ (see [23, Theorem 5.3]).

## 2.1 Null space property

As in [23], the analysis we present is based on the assumption that the measurement matrix $A$ satisfies the so-called null space property [21, 36], which is a key concept in the compressed sensing literature (see, e.g., [32, Chapter 4] for an overview).

**Definition 2.2.** *A matrix $A \in \mathbb{R}^{m \times N}$ is said to satisfy the $\ell_1$-null space property ($\ell_1$-NSP) of order $s \in \mathbb{N}$ with constant $0 < \rho_s < 1$ if for any set $S \subset [N]$ of cardinality $|S| \leq s$, it holds that $\|v_S\|_1 \leq \rho_s \|v_{S^c}\|_1$, for all $v \in \ker(A)$.*

In [32, Chapter 4], the property of Definition 2.2 was called *stable* null space property. The importance of the null space property is due to the fact that it gives a necessary and sufficient criterion for the success of basis pursuit for sparse recovery, as the following theorem shows.

**Theorem 2.3** ([32, Theorem 4.5]). *Given a matrix $A \in \mathbb{R}^{m \times N}$, every vector $x \in \mathbb{R}^N$ such that $\|x\|_0 \leq s$ is the unique solution of $(P_1)$ with $Ax = y$ if and only if $A$ satisfies the null space property of order $s$ for some $0 < \rho_s < 1$.*

The $\ell_1$-NSP is implied by the restricted isometry property (see, e.g., [11]), which is fulfilled by a large class of random matrices with high probability. For example, this includes matrices with (sub-)gaussian entries and random partial Fourier matrices [6, 60].

## 2.2 Existing theory

A predecessor of IRLS for the sparse recovery problem $(P_1)$, and more generally, for $\ell_p$-quasinorm minimization with $0 < p \leq 1$, is the *FOCal Underdetermined System Solver* (FOCUSS) as proposed by Gorodnitsky, Rao and Kreutz-Delgado [34, 59]. Asymptotic convergence of FOCUSS to a stationary point from any initialization was claimed in [59], but the proof was not entirely accurate, as

pointed out by [17]. One limitation of FOCUSS is that, unlike in IRLS as presented in Algorithm 1, no smoothing parameter $\varepsilon$ is used, which leads to ill-conditioned linear systems.

To mitigate this, [16] proposed an IRLS method that uses smoothing parameters $\varepsilon$ (such as used in $Q_\varepsilon$ defined above) that are updated iteratively. It was observed that this leads to a better condition number for the linear systems to be solved in each step of IRLS and, furthermore, that this smoothing strategy has the advantage of finding sparser vectors if the weights of IRLS are chosen to minimize a non-convex $\ell_p$-quasinorm for $p < 1$.

Further progress for IRLS designed to minimize an $\ell_1$-norm was achieved in the seminal paper [23]. In [23], it was shown that if the measurement operator fulfills a suitable $\ell_1$-null space property as in Definition 2.2, an IRLS method with iteratively updated smoothing converges to an $s$-sparse solution, coinciding with the $\ell_1$-minimizer, if there exists one that is compatible with the measurements. This method uses not exactly the update rule of (4), but rather updates the smoothing parameter such that $\varepsilon_{k+1} = \min(\varepsilon_k, R(x^{k+1})_{s+1}/N)$, where $R(x^{k+1})_{s+1}$ is the $(s+1)^{\text{st}}$-largest element of the set $\{|x_j^{k+1}|, j \in [N]\}$. Furthermore, a *local linear convergence rate* of IRLS was established [23, Theorem 6.1] under same conditions.

However, the analysis of [23] has its limitations: First, there is a gap in the assumption of their convergence results between the sparsity $s$ of a vector to be recovered and the order $\widehat{s}$ of the NSP of the measurement operator. Recently, this gap was circumvented in [3] with an IRLS algorithm that uses a smoothing update rule based on an $\ell_1$-norm, namely, $\varepsilon_{k+1} = \min(\varepsilon_k, \eta(1 - \rho_s)\sigma_s(x^{k+1})_{\ell_1}/N)$, where $\eta \in (0, 1)$, and $\rho_s$ is the NSP constant of the order $s$ of the NSP fulfilled by the measurement matrix $A$—this rule is quite similar to the rule (4) that we use in Algorithm 1. In particular, [3, Theorem III.6] establishes convergence with local linear rate similar to [23] without the gap mentioned above. The main limitation, however, of the theory of [23] (which is shared by [3]) is that the linear convergence rate only holds *locally*, i.e., in a situation where the support of the sparse vector has already been identified, see also Section 3 and Section 4.1 for a discussion.

We finally mention three relevant papers for the theoretical understanding of IRLS. [5] established the correspondence between the IRLS algorithms and the Expectation-Maximization algorithm for constrained maximum likelihood estimation under a Gaussian scale mixture distribution. By doing so, they established similar results as those from [23], i.e., the global convergence of IRLS with local linear convergence rate (as can be seen from their equation (38), which similar to (7) below) but by using different techniques based on such correspondence. [62] explores the relationship of IRLS for $\ell_1$-minimization and a slime mold dynamics, interpreting both as an instance of the same meta-algorithm. Without requiring any connection between sparse recovery and $\ell_1$-minimization, [29] shows that an IRLS-like algorithm for $(P_1)$, requires $O(N^{1/3}\log(1/\delta)/\delta^{2/3} + \log(N)/\delta^2)$ iterations to obtain a multiplicative error of $1 + \delta$ on the minimizer $||x||_1$. Unlike our result Theorem 3.2, this corresponds not to a linear, but to a sublinear convergence rate.

## 2.3 Related work

As mentioned in the introduction, IRLS has a long history and has appeared under different names within different communities, e.g., similar algorithms are usually called *half-quadratic algorithms* in image processing [2, 41] and the *Kačanov method* in numerical PDEs [25]. Probably the most common usage of IRLS has been in robust regression [35, 39], c.f. [10] for a survey that also covers applications in approximation theory. For $p$-norm regression, [1] proposed a version of IRLS for which convergence results for $p \in [2, \infty)$ were established, solving a problem that was open for over thirty years. Also, for robust regression, by using an $\ell_1$-objective on the residual, [53] showed recently global convergence of IRLS with a linear rate, with high probability for sub-Gaussian data. We note that our proof strategy is different from the one of [53] due to a structural difference of $(P_1)$ from robust regression.

In [57], the authors provide a general framework for formulating IRLS algorithms for the optimization of a quite general class of non-convex and non-smooth functions, however, without updated smoothing. They use techniques developed in [4] to show convergence of the sequence of iterates to a critical point under the Kurdyka-Łojasiewicz property [8]. However, no results about convergence rates were presented.

For the sparse recovery problem, the topic discussed in this paper, the references [30, 44, 68] analyzed IRLS for an unconstrained version of $(P_1)$, which is usually a preferable formulation if

the measurements are corrupted by noise. Additionally, the work [30] addressed the question of how to solve the successive quadratic optimization problems. The authors developed a theory that shows, under the NSP, how accurately the quadratic subproblems need to be solved via the conjugate gradient method in order to preserve the convergence results established in [23].

Finally, for the related problems of low-rank matrix recovery and completion, IRLS strategies have emerged as one of the most successful methods in terms of data-efficiency and scalability [31, 42, 43, 51].

While we were writing this paper, the manuscript [58] appeared providing new insights about IRLS. It describes a surprisingly simple reparametrization of the IRLS formulation for $\ell_p$-minimization (with $p \in (2/3, 1)$) that leads to a smooth bilevel optimization problem without any spurious minima, i.e., the stationary points of this new formulation are either global minima or strict saddles. It is an interesting future direction to explore the connection between this new approach and our global convergence theory.

## 3  IRLS for Basis Pursuit with Global Linear Rate

As discussed in Section 2.2, the main theoretical advancements for IRLS for the sparse recovery problem were achieved in the work [23].

**Proposition 3.1.** [23, Theorem 6.1] *Assume that $A \in \mathbb{R}^{m \times N}$ satisfies the NSP of order $\widehat{s} > s$ with constant $\rho_{\widehat{s}}$ such that $0 < \rho_{\widehat{s}} < 1 - \frac{2}{\widehat{s}+2}$ and $\widehat{s} > s + \frac{2\rho_{\widehat{s}}}{1-\rho_{\widehat{s}}}$ hold. Let $x_* \in \mathbb{R}^N$ be an $s$-sparse vector and set $y = Ax_*$. Assume that there exists an integer $k_0 \geq 1$ and a positive number $\xi > 0$ such that*

$$\xi := \frac{\|x^{k_0} - x_*\|_1}{\min_{i \in S} |(x_*)_i|} < 1. \tag{7}$$

*Then the iterates $\{x^{k_0}, x^{k_0+1}, x^{k_0+2}, \ldots\}$ of the IRLS method in [23] converge linearly to $x_*$, i.e., for all $k \geq k_0$, the $k$th iteration of IRLS satisfies*

$$\|x^{k+1} - x_*\|_1 \leq \frac{\rho_{\widehat{s}}(1 + \rho_{\widehat{s}})}{1 - \xi} \left( 1 + \frac{1}{\widehat{s} - 1 - s} \right) \|x^k - x_*\|_1. \tag{8}$$

The main contribution of this paper is that we overcome a local assumption such as (7) and show that IRLS as defined by Algorithm 1 *exhibits a global linear convergence rate*, i.e., there is a linear convergence rate starting from any initialization, as early as in the first iteration.

**Exactly sparse case.** Our first main result, Theorem 3.2, deals with the scenario that the ground truth vector $x_*$ is exactly $s$-sparse. Our second result, presented in the supplementary material, generalizes the first one to the more realistic situation of approximately sparse vectors.

**Theorem 3.2.** *Consider the problem of recovering an unknown $s$-sparse vector $x_* \in \mathbb{R}^N$ from known measurements of the form $y = Ax_*$. Assume that the measurement matrix $A \in \mathbb{R}^{m \times N}$ fulfills the $\ell_1$-NSP of order $s$ with constant $\rho_s < 1/2$. Let the IRLS iterates $\{x^k\}_k$ and $\{\varepsilon_k\}_k$ be defined by the IRLS algorithm (3) and (4) with initialization $x^0$. Then, for all $k \in \mathbb{N}$, it holds that*

$$\mathcal{J}_{\varepsilon_k}(x^k) - \|x_*\|_1 \leq \left( 1 - \frac{c}{\rho_1 N} \right)^k \left( \mathcal{J}_{\varepsilon_0}(x^0) - \|x_*\|_1 \right) \tag{9}$$

*as well as*

$$\|x^k - x_*\|_1 \leq 9 \left( 1 - \frac{c}{\rho_1 N} \right)^k \|x^0 - x_*\|_1. \tag{10}$$

*Here $c = 1/768$ is an absolute constant and $\rho_1 < 1/2$ denotes the $\ell_1$-NSP constant of order $1$.*

Inequality (9) says that the difference $\mathcal{J}_{\varepsilon_k}(x^k) - \|x_*\|_1$ converges linearly with a uniform upper bound of $1 - \frac{c}{\rho_1 N}$ on the linear convergence factor. As our proof, which is detailed in the supplementary material, shows, this implies inequality (10), which implies that also $\|x_* - x^k\|_1$ exhibits linear convergence in the number of iterations $k$. In particular, this means that for some error tolerance $\delta > 0$, we obtain $\|x_* - x^k\|_1 \leq \delta$ after $O\left( \rho_1 N \log \left( \frac{\|x_* - x^0\|_1}{\delta} \right) \right)$ iterations.

**Remark 3.3.** *Note that it follows directly from Definition 2.2 that the constant $\rho_1$ of the $\ell_1$-NSP of order 1 satisfies $\rho_1 \leq \rho_s \leq 1$, which implies that $\delta$-accuracy is obtained after $O\left(N \log\left(\frac{\|x_* - x^0\|_1}{\delta}\right)\right)$ iterations. This bound can be improved in many scenarios where one can obtain more explicit bounds on $\rho_1$, for example, when $A$ is a Gaussian matrix. Namely, inspecting [32, p. 142 and Thm. 9.2], we observe in this scenario that $\rho_1 \lesssim \sqrt{(\log N)/m}$ with high probability. Hence, in this scenario, at most $O\left(N\sqrt{\frac{\log N}{m}} \log\left(\frac{\|x_* - x^0\|_1}{\delta}\right)\right)$ iterations are needed to achieve $\delta$-accuracy.*

The key idea in our proof is to use fact that the quadratic functional $Q_{\varepsilon_k}(\cdot, x^k)$ approximates the $\ell_1$-norm in a neighborhood of the current iterate $x^k$. For this reason, we also expect that for $t > 0$ sufficiently small, we have that $Q_{\varepsilon_k}(x^k + tv^k, x^k) < Q_{\varepsilon_k}(x^k, x^k)$ if $v^k = x_* - x^k$ is the vector between $x^k$ and the ground truth $x_*$. Then by choosing $t$ properly, we can guarantee a sufficient decrease of the functional $\mathcal{J}_{\varepsilon_k}\left(x^k\right)$ in each iteration.

In Section 4, we conduct experiments that indeed verify the linear convergence in (9) and (10). Moreover, we study numerically whether one can observe a dependence of the convergence rate on the problem parameters $N$, $s$ and $m$. We construct a worst-case example which indicates that the convergence rate indeed may depend on the dimension $N$ in a way as described by (9). In a certain sense, this indicates that there are two convergence phases, a global one, where a dimension-dependent constant cannot be avoided and a local convergence phase, where a local convergence result such as described in Proposition 3.1 kicks in.

Finally, let us mention that we have undertaken no efforts to optimize the constant $c = 1/768$ in Theorem 3.2. Nevertheless, we note that the constant $c$ can be replaced by the sharper constant $c_{\rho_s}$ as defined in Proposition B.3.

## 4 Numerical experiments

In this section, we support our theory with numerical experiments. First, we examine whether IRLS indeed exhibits two distinct convergence phases, a "global" one, as described in this paper, and a local one, as described in [3, 23], corresponding to different linear convergence rate factors. Second, we explore to which extent the dimension dependence in the convergence rates (9) and (10) indicated by Theorem 3.2 is necessary, or if we rather can expect a dimension-free linear convergence rate factor. All experiments are conducted on an iMac computer with a 4 GHz Quad-Core Intel Core i7 CPU, using MATLAB R2020b.

### 4.1 Local and global convergence phase

We first note that the local convergence result of [23, Theorem 6.1] depends on the locality condition $\xi(k) := \frac{\|x^k - x_*\|_1}{\min_{i \in S} |(x_*)_i|} < 1$, cf. (7). Under this condition (and an appropriate null space condition), Proposition 3.1 stated above implies that $\|x^{k+1} - x_*\|_1 \leq \mu \|x^k - x_*\|_1$ with an absolute constant $\mu < 1$ which, in particular, does *not* depend on the dimension $N, m$, and $s$. This corresponds to a locally linear rate for IRLS. A very similar condition to (7) is required by the comparable and more recent local convergence statement [3, Theorem III.6, inequality (III.14)] for the IRLS variant considered in [3].

However, a closer look at the locality condition (7) reveal that its *basin of attraction* is very restrictive: This condition means that the *support identification* problem underlying the sparse recovery *has already been solved*, as can be seen from the following proposition, whose proof we provide in the supplementary material.

**Proposition 4.1.** *Let $x^k, x_* \in \mathbb{R}^N$, let $S \subset [N]$ be the support set of $x_*$ of size $|S| = s$. If (7) holds, i.e., if $\|x^k - x_*\|_1 < \min_{i \in S} |(x_*)_i|$, then the set $S_k \subset [N]$ of the $s$ largest coordinates of $x^k$ coincides with $S$.*

We now explore the behavior of the IRLS algorithm for $\ell_1$-minimization, Algorithm 1, and the sharpness of Proposition 4.1 in experiments that build on those of [23, Section 8.1]. For this purpose, for $N = 8000$, we sample independently a 200-sparse vector $x_* \in \mathbb{R}^N$ with random support $S \subset [N]$, $s = 200 = |S|$, chosen uniformly at random such that $(x_*)_S$ is chosen according the Haar measure

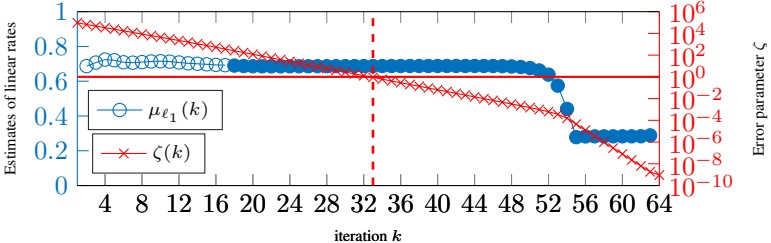

Figure 1: Instantaneous linear convergence rates of IRLS for $\ell_1$-minimization ($N = 8000$): Linear convergence factors $\mu_{\ell_1}(k) := \|x^k - x_*\|_1 / \|x^{k-1} - x_*\|_1$ (in blue), filled blue circle if $S_k = S$ with $S_k$ of Proposition 4.1 (support identification), and error parameter $\zeta(k) := \|x^k - x_*\|_1 / \min_{i \in S} |(x_*)_i|$ (in red). Horizontal (red) line: Threshold $\zeta = 1$. Vertical (red) line: First iterate $k$ with $\zeta(k) < 1$.

on the sphere of a 200-dimensional unit $\ell_2$-ball, and a measurement matrix $A \in \mathbb{R}^{m \times N}$ with i.i.d. Gaussian entries such that $A_{ij} \sim \mathcal{N}(0, 1/m)$, while setting $m = \lfloor 2s \log(N/s) \rfloor$. Such a matrix is known to fulfill with high probability the $\ell_1$-null space property of order $s$ with constant $\rho_s < 1$ [32, Theorem 9.29].

In Figure 1, we track the decay of the $\ell_1$-error $\|x^k - x_*\|_1$ of the iterates $x^k$ returned by Algorithm 1 via the values of $\zeta(k) := \|x^k - x_*\|_1 / \min_{i \in S} |(x_*)_i|$, depicted in red, and the behavior of the factor $\mu_{\ell_1}(k) := \|x^k - x_*\|_1 / \|x^{k-1} - x_*\|_1$, depicted in blue. We observe that the condition (7) for local convergence with the fast, dimension-less linear rate (8) is satisfied after $k = 33$ iterations, as indicated by the vertical dashed red line.

In the first few iterations, $\zeta(k)$ is larger than 1 by several orders of magnitudes, suggesting that the local convergence rate results of [3, 23] do *not* apply until the later stages of the simulation: In fact, we observe that the support $S$ of $x_*$ is already perfectly identified via the $s$ largest coordinates of $x^k$ as soon as $k \geq 18$. For iterations $18 \leq k \leq 50$, the linear rate $\mu_{\ell_1}(k)$ remains very stably around $\approx 0.7$, after which an accelerated linear rate can be observed.[5] Before $k = 18$, for this example, the rate $\mu(k)$ hovers around $0.7$ with slight variations. For all iterations $k$, $\mu(k)$ is smaller than 1, in line with the global linear convergence rate implied by Theorem 3.2.

Repeating a similar experiment for a larger ambient space dimension $N = 16000$ and a smaller measurement-to-sparsity ratio such that $m = \lfloor 1.75s \log(N/s) \rfloor$ results in a qualitatively similar situation, as seen in Figure 2(a): In Figure 2(a), we add also a plot of the linear convergence factor $\mu(k) := \frac{\mathcal{J}_{\varepsilon_k}(x^k) - \|x_*\|_1}{\mathcal{J}_{\varepsilon_{k-1}}(x^{k-1}) - \|x_*\|_1}$ that tracks the behavior of the linear convergences in the smoothed $\ell_1$-norm objective $\mathcal{J}$, cf. (17). In addition to what have been observed in Figure 1, we see that $\mu(k)$ and $\mu_{\ell_1}(k)$ exhibit a very similar behavior for this example.

Hence, these experiments indicate that we can distinguish two phases. In the first, global phase linear convergence already sets in, but the instantaneous linear convergence rate has not yet stabilized. In the second one, when the support identification problem has been solved, the instantaneous linear convergence stabilizes.

**Remark 4.2.** *There are other methods in the literature, such as proximal algorithms, for which convergence results with a two phase behaviour were already established. For example, [47] showed that a forward-backward method applied to the Lasso problem exhibits local linear convergence, and that after a finite number of iterations, the region of fast convergence is reached. In particular, [47, Proposition 3.6(ii)] provides a bound on this number of iterations, which scales proportionally with $\|x_* - x^0\|_2^2$. On the other hand, (3.2) for IRLS provides a bound on the number of iterations until the fast linear convergence rate is reached that scales proportionally with $\log(\|x_* - x^0\|_2)$, but also proportionally with the dimension $N$. Moreover, most of these results require stronger assumptions than the NSP, such as the restricted isometry property or a restricted strong convexity/smoothness property.*

---

[5]The latter phenomenon cannot be observed for the IRLS algorithm of [23] as it uses a slightly different objective function than Algorithm 1.

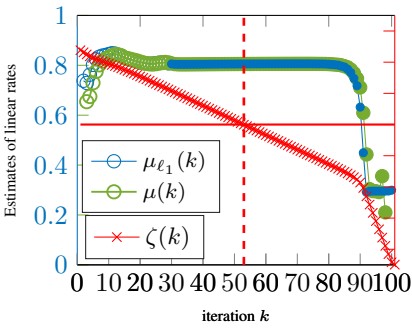

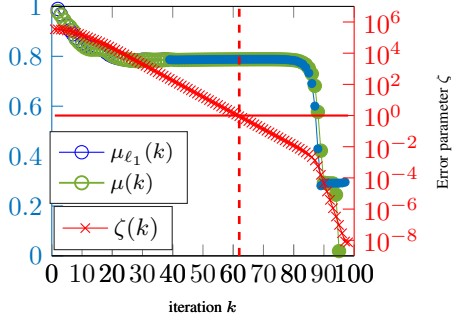

(a) Standard initialization (uniform weights $(w_0)_i = 1$ for all $i$).

(b) Adversary initialization (weights $(w_0)_i$ as in (11)).

Figure 2: Instantaneous linear convergence rates of IRLS for $\ell_1$-minimization ($N = 16000$): Linear convergence factors $\mu_{\ell_1}(k) := \frac{\|x^k - x_*\|_1}{\|x^{k-1} - x_*\|_1}$ (in blue) and $\mu(k) := \frac{\mathcal{J}_{\varepsilon_k}(x^k) - \|x_*\|_1}{\mathcal{J}_{\varepsilon_{k-1}}(x^{k-1}) - \|x_*\|_1}$ (in green), filled circles if $S_k = S$ (perfect support identification), and error parameter $\zeta(k) := \|x^k - x_*\|_1 / \min_{i \in S} |(x_*)_i|$ (in red), horizontal and vertical red lines as in Figure 1.

## 4.2 Global convergence rate and its dimension dependence

In this section, we explore to which extent the dependence on $N$ in the convergence rates (10) and (14) is necessary or if we can rather expect a dimension-free linear convergence rate factor. To this end, we run a variation of IRLS that initializes the weight vector $w_0 \in \mathbb{R}^N$ not uniformly as in Algorithm 1, but based on an *adversary initialization*, here denoted by $z^{\mathrm{adv}}$. More specifically, we first compute a minimizer

$$z^{\mathrm{adv}} \in \underset{z \in \mathbb{R}^{S^c} : A_{S^c} z = y}{\arg\min} \|z\|_1$$

of the $\ell_1$-minimization problem restricted to the off-support coordinates of $x_*$ indexed by $S^c$ and set then $x^0 \in \mathbb{R}^N$ such that $x^0_{S^c} := z^{\mathrm{adv}}$ and $x^0_S = 0$. Based on this *initialization* $x^0$, we compute $\varepsilon_0 := \frac{\sigma_s(x^0)_{\ell_1}}{N}$ and set the first weight vector such that for all $i \in [N]$,

$$(w_0)_i := \frac{1}{\max(|x^0_i|, \varepsilon_0)}, \tag{11}$$

before proceeding with the IRLS steps (3), (4) and (5) until convergence.

We observe in Figure 2(b) that this initialization, which is *adversary* as it sets very large initial weights on the coordinates of $S$ that correspond to the true support of $x_*$, eventually results in the same behavior of Algorithm 1 as for the standard initialization by uniform weights, identifying the true support at iteration $k = 39$ compared to $k = 30$. However, in the first few iterations, we see that the instantaneous linear convergence factor $\mu(k)$ is close to 1 with $\mu(1) = 0.980$, decreasing only slowly before stabilizing around 0.79 after around $k = 30$.

While this is just one example, this already indicates that in general, a linear rate such as (8), i.e., without dependence on the dimension $N$ (which has been proven locally in [23, Theorem 6.1] and [3, Theorem III.6]) might not hold in general.

In our next experiment, we further investigate numerically the dimension dependence of the worst-case linear convergence factor $\mu(k) := \frac{\mathcal{J}_{\varepsilon_k}(x^k) - \|x_*\|_1}{\mathcal{J}_{\varepsilon_{k-1}}(x^{k-1}) - \|x_*\|_1}$, which is upper bounded by the result of Theorem 3.2. We saw that in the experiment using the adversary initialization mentioned above and depicted in Figure 2(b), the maximal value was attained in the first iteration, i.e., for $\mu(1)$, as the effect of the adversary initialization is most eminent for $k = 1$.

We now run IRLS starting from the adversary initialization for different ambient dimensions $N = 125 \cdot 2^{\ell/2}$ for $\ell = 0, 1, \ldots, 14$. For each of the values of $N$, we sample vectors $x_* \in \mathbb{R}^N$ of sparsity $s = 40$ from the same random model as above, and scale the number of i.i.d. Gaussian measurements with $m = \lfloor 2s \log(N/s) \rfloor$. We average the resulting values for $\mu(1)$ across 500 independent realizations of the experiment.

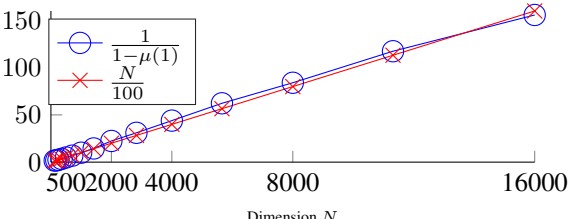

Figure 3: Comparison of $\frac{N}{100}$ and $\frac{1}{1-\mu(1)}$ (for which Proposition B.3 provides un upper bound of $\frac{\rho_1 N}{c}$) for different dimension parameters $N$, where $\mu(1) = \frac{\mathcal{J}_{\varepsilon_1}(x^1) - \|x_*\|_1}{\mathcal{J}_{\varepsilon_0}(x^0) - \|x_*\|_1}$ is the linear convergence factor, for IRLS initialized from adversary initialization.

In Figure 3, we see that dependence on $N$ of linear convergence factor $\mu(1)$ that is observed for this experiment is quite well described by the upper bound (9) provided by our main result Theorem 3.2, as $\frac{1}{1-\mu(1)}$ scales almost linearly with $N$. As a footnote in Section 3 indicates, the constant $\rho_1$ of the null space property of order 1 scales with $\sqrt{\frac{\log N}{m}}$, and therefore a precise dependence on all the parameters such as $m$ and $s$ might be more complicated than what can be observed in this experiment.

Nevertheless, we interpret Figure 3 as strong evidence that the linear convergence rate factor of Proposition B.3 is tight in its dependence on $N$, and that a dimension-less factor $\mu$ cannot be expected in general.

## Conclusion

In this paper, we solved an open problem in the algorithmic theory for sparse recovery. In particular, we established a new variant of the IRLS algorithm for Basis Pursuit or $\ell_1$-minimization for which we show a global linear convergence rate under a suitable and sharp assumption, namely, the null-space property. Moreover, we have corroborated our theory with numerical experiments that, first, discussed the difference between the local and global convergence phase and, second, that elucidated the optimality of the dimension dependence of convergence rate given by our main theorem.

We think that the results in this paper give rise to a number of interesting research directions for follow-up work. While the numerical experiments in Section 4 substantiate the hypothesis that the dependence of the convergence rate on $N$ and $\rho_1$ in our theory is not an artifact of our proof, we also observed in this section that for a *generic initialization* no such dependence can be observed. In view of this, it is interesting to investigate whether a dimension-independent global convergence rate is possible, for example via a *smoothed analysis* [22, 61]. Furthermore, there are currently no convergence rates available for IRLS optimizing a nuclear norm-type objective, which is of great interest for low-rank matrix recovery [31, 43, 51], and we expect that our analysis may be generalizable to this setting as well.

Finally, it was observed that sparse vectors can be recovered from even fewer measurements via the optimization of a non-convex $\ell_p$-quasinorm (with $0 < p < 1$), and that IRLS exhibits excellent performance in this case [16, 23]. While a thorough understanding has remained elusive so far for this non-convex case, we consider our results as a first step towards a global convergence theory for IRLS for the optimization of $\ell_p$-quasinorms or similar non-convex surrogate objectives.

## Acknowledgments and Disclosure of Funding

The authors want to thank Massimo Fornasier for inspiring discussions and the anonymous reviewers for their detailed comments. C.K. is grateful for support by the NSF under the grant NSF-IIS-1837991. C.M.V. is supported by the German Federal Ministry of Education and Research (BMBF) in the context of the Collaborative Research Project SparseMRI3D+ and by the German Science Foundation (DFG) in the context of the project KR 4512/1-1. D.S. was supported by the Air Force Office of Scientific Research under award #FA9550-18-1-00788.

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
