In this supplement to the paper, we present in Section A our second main result, Theorem A.1, which generalizes the global linear convergence of Algorithm 1 for sparse vectors, which we presented Theorem 3.2, to approximately sparse vectors. In Section B we provide the proof of our theoretical results. In Section C, we discuss how to solve the (weighted) least squares step in Algorithm 1 in a computationally efficient manner. Finally, Section D contains, for completeness, a proof of two technical results, namely Lemma 2.1 and Proposition 4.1.

## Contents

## A  Linear Convergence of IRLS for approximately sparse vectors

We now generalize Theorem 3.2 to the scenario where the ground truth $x_*$ is only approximately sparse. By that, we mean that the vector $x_*$ can be well-approximated by an $s$-sparse vector in the sense that the $\ell_1$-error of the best $s$-term approximation $\sigma_s(x_*)_{\ell_1} = \inf\{\|x_* - z\|_1 : z \in \mathbb{R}^N \text{ is } s\text{-sparse}\}$ is small, which is a commonly used quantity to measure the model misfit to a sparse vector [32, Section 2.1]. If $x_*$ is approximately sparse in this sense, we can only hope to *approximately* recover $x_*$ by the $\ell_1$-minimization program $(P_1)$. Indeed, [23, Theorem 5.3(iv)] showed that under a suitable null space property, IRLS for $(P_1)$ finds a vector $x$, such that $\|x - x_*\|_1$ is at most a constant multiple of the optimal best $s$-term approximation error $\sigma_s(x_*)_{\ell_1}$. However, as for exactly sparse vectors $x_*$, only a local, but no global convergence rate was provided in previous literature [23, Theorem 6.4].

The following result shows that in fact, we also can obtain global linear convergence of Algorithm 1 in this case. More precisely, Theorem A.1 implies that $\mathcal{J}_{\varepsilon_k}(x^k) - \|x_*\|_1$ decays exponentially fast until a certain accuracy is reached, which is $\sigma_s(x_*)_{\ell_1}$ up to a constant multiple.

**Theorem A.1.** *Consider the problem of recovering an unknown vector $x_* \in \mathbb{R}^N$ from known measurements of the form $y = Ax_*$. Assume that the measurement matrix $A \in \mathbb{R}^{m \times N}$ fulfills the $\ell_1$-NSP of order $s$ with constant $\rho_s < 1/8$. Let the IRLS iterates $\{x^k\}_k$ and $\{\varepsilon_k\}_k$ be defined by (3) and (4) with initialization $x^0$. Then the following three statements hold.*

1. *For $k \leq \hat{k} := \min\left\{k \in \mathbb{N} : \sigma_s(x_*)_{\ell_1} > \frac{2}{9}\left\|(x_* - x^k)_{S^c}\right\|_1\right\}$ it holds that*

$$\mathcal{J}_{\varepsilon_k}(x^k) - \|x_*\|_1 \leq \left(1 - \frac{c}{\rho_1 N}\right)^k \left(\mathcal{J}_{\varepsilon_0}(x^0) - \|x_*\|_1\right), \qquad (12)$$

   *where $S$ denotes the support of the $s$ largest entries of $x_*$.*

2. *For all $1 \leq k \leq \hat{k}$ it holds that*

$$\|x^k - x_*\|_1 \leq 6\left(1 - \frac{c}{\rho_1 N}\right)^k \|x^0 - x_*\|_1 + 10\sigma_s(x_*)_{\ell_1}. \qquad (13)$$

3. *Moreover, for all integers $k \gtrsim \rho_1 N \log \left( \frac{\|x^0 - x_*\|_1}{\sigma_s (x_*)_{\ell_1}} \right)$ we have that*

$$\|x^k - x_*\|_1 \leq 20\sigma_s (x_*)_{\ell_1} . \tag{14}$$

*Here $c = 1/3072$ and $\rho_1 < 1/8$ denotes the constant for the $\ell_1$-NSP of order 1.*

**Remark A.2.** *Applying Theorem A.1 to the special case $\sigma_s (x_*)_{\ell_1} = 0$, we observe that inequality (10) yields a seemingly sharper result than inequality (13) in Theorem 3.2, which may seem somewhat counterintuitive. However, note that in Theorem 3.2 we require $\rho_s < 1/2$, whereas in Theorem A.1 we have the stronger assumption $\rho_s < 1/8$. Indeed, a closer inspection of the proofs reveals that both the factors 3 and 6 in the inequalities (10) and (13) can be replaced by the factor $\frac{3(1+\rho_s)}{1-\rho_s}$, reconciling those two results.*

## B   Proofs of the main results

In this section, we prove the main results of this paper, Theorem 3.2, and its approximately sparse counterpart, Theorem A.1. To this end, we first state and prove the following technical lemma, which gives an upper and lower bound for $\mathcal{J}_\varepsilon (x) - \|x_*\|_1$, which is the quantity for which we are going to show linear convergence.

**Lemma B.1.** *Let $x_*, x \in \mathbb{R}^N$. Assume that $A$ fulfills the $\ell_1$-NSP of order $s$ with constant $\rho_s < 1$. Furthermore, suppose $Ax_* = Ax$ and that $\varepsilon \leq \frac{1}{N}\sigma_s (x)_{\ell_1}$. Then it holds that*

$$\frac{1 - \rho_s}{1 + \rho_s}\|x - x_*\|_1 - 2\sigma_s (x_*)_{\ell_1} \leq \mathcal{J}_\varepsilon (x) - \|x_*\|_1 \leq 3\sigma_s (x)_{\ell_1} . \tag{15}$$

In order to prove Lemma B.1 we need the following technical lemma.

**Lemma B.2.** [23, Lemma 4.3] *Assume that the matrix $A \in \mathbb{R}^{m \times N}$ fulfills the $\ell_1$-NSP for some $s$ and $\rho_s < 1$. Then for all $x, x_* \in \mathbb{R}^N$ such that $Ax = Ax_*$ it holds that*

$$\|x - x_*\|_{\ell_1} \leqslant \frac{1 + \rho_s}{1 - \rho_s} \left(\|x_*\|_1 - \|x\|_1 + 2\sigma_s(x)_{\ell_1}\right) .$$

*Proof of Lemma B.1.* We observe that $\mathcal{J}_\varepsilon (x) \geq \|x\|_1$ for each $x \in \mathbb{R}^N$, which follows directly from the definition of $\mathcal{J}_\varepsilon (x)$, see Equation (1). Now, fix $x_*, x \in \mathbb{R}^N$, and let $S$ be the set which contains the $s$ largest entries of $x$ in absolute value. Hence, we obtain that

$$
\begin{aligned}
\mathcal{J}_\varepsilon (x) - \|x_*\|_1 &\geq \|x\|_1 - \|x_*\|_1 \\
&= \|x_{S^c}\|_1 + \|x_S\|_1 - \|x_*\|_1 \\
&\geq \|x_{S^c}\|_1 - \|(x - x_*)_S\|_1 - \|(x_*)_{S^c}\|_1 \\
&\geq \|(x - x_*)_{S^c}\|_1 - \|(x - x_*)_S\|_1 - 2\|(x_*)_{S^c}\|_1,
\end{aligned}
$$

where in each of the last two inequalities we have applied the reverse triangle inequality. Since $x - x_*$ is contained in the null space of $A$, it follows from the nullspace property that $\|(x - x_*)_S\|_1 \leq \rho_s\|(x - x_*)_{S^c}\|_1$. Hence, we have shown that

$$\mathcal{J}_\varepsilon (x) - \|x_*\|_1 \geq (1 - \rho_s)\|(x - x_*)_{S^c}\|_1 - 2\|(x_*)_{S^c}\|_1.$$

Since it follows from the null space property that $\|(x - x_*)_{S^c}\|_1 \geq \frac{\|x - x_*\|_1}{1+\rho_s}$, this shows the first inequality in (15).

Next, we are going to prove the reverse inequality in (15). For that, set $I := \{i \in [N] : |x_i| > \varepsilon\}$. Then we observe that

$$
\begin{aligned}
\mathcal{J}_\varepsilon (x) - \|x_*\|_1 = \|x_I\|_1 + \frac{1}{2} \sum_{i \in I^c} \left(\frac{x_i^2}{\varepsilon} + \varepsilon\right) - \|x_*\|_1 \\
\leq \|x_I\|_1 + |I^c|\varepsilon - \|x_*\|_1 \\
\leq \|x_I\|_1 + \sigma_s (x)_{\ell_1} - \|x_*\|_1 \\
\leq \|x\|_1 + \sigma_s (x)_{\ell_1} - \|x_*\|_1 .
\end{aligned}
\tag{16}
$$

In the third line we used the assumption $\varepsilon \leq \frac{1}{N}\sigma_s(x)_{\ell_1}$. In order to proceed, we first derive an appropriate upper bound for $\|x\|_1 - \|x_*\|_1$. For that, we note

$$\left(\frac{1-\rho_s}{1+\rho_s} + 1\right)(\|x\|_1 - \|x_*\|_1) \leq \frac{1-\rho_s}{1+\rho_s}\|x - x_*\|_1 - (\|x_*\|_1 - \|x\|_1)$$

$$\leq \left(\|x_*\|_1 - \|x\|_1 + 2\sigma_s(x)_{\ell_1}\right) - (\|x_*\|_1 - \|x\|_1)$$

$$\leq 2\sigma_s(x)_{\ell_1},$$

where in the second line we have used Lemma B.2. This shows that $\|x\|_1 - \|x_*\|_1 \leq \frac{2\sigma_s(x)_{\ell_1}}{1+\frac{1-\rho_s}{1+\rho_s}}$. Combining this with (16), we obtain

$$\mathcal{J}_\varepsilon(x) - \|x_*\|_1 \leq 3\sigma_s(x)_{\ell_1},$$

which finishes the proof of inequality (15). $\qquad\square$

The next key proposition states that the quantity $\mathcal{J}_{\varepsilon_k}(x^k) - \|x_*\|_1$ decays linearly under appropriate conditions.

**Proposition B.3.** *Let $x_* \in \mathbb{R}^N$ be an approximately $s$-sparse vector with support $S$. Let $A \in \mathbb{R}^{m \times N}$ and $y = Ax_*$. Assume that $A$ fulfills the $\ell_1$-NSP of order $s$ with constant $\rho_s < 3/4$, if $\sigma_s(x_*)_{\ell_1} = 0$, and $\rho_s < 1/4$ otherwise. Denote by $\rho_1$ the NSP constant of order 1.*
*Let the IRLS iterates $\{x^k\}_k$ and $\{\varepsilon_k\}_k$ be defined by (3) and (4) with initialization $x^0$. Then, for all $k \in \mathbb{N}$, such that $\|(x_*)_{S^c}\|_1 \leq \frac{2}{9}\|(x_*)_{S^c} - x^\ell_{S^c}\|_1$ for all $\ell < k$, the following holds*

$$\mathcal{J}_{\varepsilon_k}(x^k) - \|x_*\|_1 \leq \left(1 - \frac{c_{\rho_s}}{\rho_1 N}\right)^k \left(\mathcal{J}_{\varepsilon_0}(x^0) - \|x_*\|_1\right). \tag{17}$$

*where the constant $c_{\rho_s}$ is defined by*

$$c_{\rho_s} := \begin{cases} \frac{(3/4-\rho_s)^2}{48} & \text{if } \sigma_s(x_*)_{\ell_1} = 0 \\ \frac{(1/4-\rho_s)^2}{48} & \text{else} \end{cases}$$

Before proving this statement, let us describe the main ideas of our proof. Recall that $x_*$ has minimal $\|\cdot\|_1$-norm among all vectors $x$, which satisfy the constraint $Ax = y$. Hence, setting $v^k = x_* - x^k$ due to convexity of the $\ell_1$-norm we that $\|x^k + tv^k\|_1 < \|x^k\|_1$ for all $0 < t < 1$. Since that the quadratic functional $Q(\cdot, x^k)$ approximates the objective function $\mathcal{J}_\varepsilon$, which is a surrogate for the $\ell_1$-norm, in a neighborhood of the current iterate $x^k$, we also expect that for $t > 0$ sufficiently small we have that $Q(x^k + tv^k, x^k) < Q(x^k, x^k)$. In order to show that the decrease is sufficiently large, we also need to show that $t$ can be chosen large enough. This will guarantee a sufficient decrease of $\mathcal{J}_{\varepsilon_k}(x^k)$ in each iteration.

*Proof of Proposition B.3.* In order to show inequality (17) we will prove by induction that for each $k$, such that $\|(x_*)_{S^c}\|_1 \leq \frac{2}{9}\|(x_*)_{S^c} - x^\ell_{S^c}\|_1$ for all $\ell \leq k$, it holds that

$$\mathcal{J}_{\varepsilon_{k+1}}(x^{k+1}) - \|x_*\|_1 \leq \left(1 - \frac{c_{\rho_s}}{\rho_1 N}\right)\left(\mathcal{J}_{\varepsilon_k}(x^k) - \|x_*\|_1\right).$$

Now choose such a $k \geq 1$ and assume that the statement has been shown for all $k' < k$. Set $v^k = x_* - x^k$. For $t \in \mathbb{R}$, we have, by optimality of $x^{k+1}$ in (3), that

$$\mathcal{J}_{\varepsilon_{k+1}}(x^{k+1}) \leq Q_{\varepsilon_k}(x^{k+1}, x^k) \leq Q_{\varepsilon_k}(x^k + tv^k, x^k). \tag{18}$$

Moreover, by the definition of the quadratic objective $Q_{\varepsilon_k}(\cdot, x^k)$ (see (2)), it holds that

$$Q_{\varepsilon_k}(x^k + tv^k, x^k) - \mathcal{J}_{\varepsilon_k}(x^k) = t\langle\nabla\mathcal{J}_{\varepsilon_k}(x^k), v^k\rangle + \frac{t^2}{2}\langle v^k, \mathrm{diag}(w_{\varepsilon_k}(x^k))v^k\rangle. \tag{19}$$

Our goal is to show that by picking $t$ large enough, we can make $Q_{\varepsilon_k}(x^k + tv^k, x^k) - \mathcal{J}_{\varepsilon_k}(x^k) < 0$ sufficiently small. For that, we now control $\langle\nabla\mathcal{J}_{\varepsilon_k}(x^k), v^k\rangle$ and $\langle v^k, \mathrm{diag}(w_{\varepsilon_k}(x^k))v^k\rangle$ separately.

**Part I: Bounding the linear term $\langle \nabla \mathcal{J}_{\varepsilon_k}(x^k), v^k \rangle$:**

Let $I := \{i \in [N] : |x_i^k| > \varepsilon_k\}$ and denote by $S$ the set which contains the $s$ largest entries of $x_*$ in absolute value. In the case that $x_*$ is sparse, $S$ is given by the support of $x_*$, i.e. $S = \operatorname{supp}(x_*)$. Consider

$$\langle \nabla \mathcal{J}_{\varepsilon_k}(x^k), v^k \rangle = \sum_{i=1}^{N} \frac{x_i^k}{\max(|x_i^k|, \varepsilon_k)} v_i^k = \sum_{i \in S} \frac{x_i^k}{\max(|x_i^k|, \varepsilon_k)} v_i^k + \sum_{i \in S^c} \frac{x_i^k}{\max(|x_i^k|, \varepsilon_k)} v_i^k.$$

The first summand can be bounded by

$$\sum_{i \in S} \frac{x_i^k}{\max(|x_i^k|, \varepsilon_k)} v_i^k = \sum_{i \in S \cap I} \operatorname{sgn}(x_i^{(k)}) v_i^k + \sum_{i \in S \cap I^c} \frac{x_i^k}{\varepsilon_k} v_i^k$$
$$\leq \|v_{S \cap I}^k\|_1 + \|v_{S \cap I^c}^k\|_1$$
$$= \|v_S^k\|_1$$
$$\leq \rho_s \|v_{S^c}^k\|_1.$$

Where the last inequality comes from the NSP for $v^k$ which, in turns, comes from the fact that $v^k = x_* - x^k \in \ker(A)$, since $Ax_* = Ax^k$. For the second summand we have that

$$\sum_{i \in S^c} \frac{x_i^k}{\max(|x_i^k|, \varepsilon_k)} v_i^k$$
$$= \sum_{i \in S^c \cap I} \operatorname{sgn}(x_i^k) v_i^k + \sum_{i \in S^c \cap I^c} \frac{x_i^k v_i^k}{\varepsilon_k}$$
$$= \sum_{i \in S^c \cap I} \operatorname{sgn}(x_i^k)(x_*)_i - \sum_{i \in S^c \cap I} \operatorname{sgn}(x_i^k) x_i^k + \sum_{i \in S^c \cap I^c} \frac{x_i^k (x_*)_i}{\varepsilon_k} - \sum_{i \in S^c \cap I^c} \frac{(x_i^k)^2}{\varepsilon_k}$$
$$\leq \| (x_*)_{S^c \cap I} \|_1 - \|x_{S^c \cap I}^k\|_1 + \| (x_*)_{S^c \cap I^c} \|_1 - \frac{\|x_{S^c \cap I^c}^k\|_2^2}{\varepsilon_k}$$
$$= -\|x_{S^c \cap I}^k\|_1 + \| (x_*)_{S^c} \|_1 - \frac{\|x_{S^c \cap I^c}^k\|_2^2}{\varepsilon_k}$$
$$= \| (x_*)_{S^c} \|_1 - \|x_{S^c}^k\|_1 + \|x_{S^c \cap I^c}^k\|_1 - \frac{\|x_{S^c \cap I^c}^k\|_2^2}{\varepsilon_k}$$
$$\leq 2\| (x_*)_{S^c} \|_1 - \|v_{S^c}^k\|_1 + \|x_{S^c \cap I^c}^k\|_1 - \frac{\|x_{S^c \cap I^c}^k\|_2^2}{\varepsilon_k}.$$

In order to proceed, we note that from the elementary inequality $ab \leq \frac{1}{2}(a^2 + b^2)$ and from $\|x_{S^c \cap I^c}^k\|_1 \leq \sqrt{N}\|x_{S^c \cap I^c}^k\|_2$, it follows that

$$\|x_{S^c \cap I^c}^k\|_1 \leq \frac{1}{2}\left(\frac{\varepsilon_k \|x_{S^c \cap I^c}^k\|_1^2}{2\|x_{S^c \cap I^c}^k\|_2^2} + 2\frac{\|x_{S^c \cap I^c}^k\|_2^2}{\varepsilon_k}.\right) \leq \frac{\varepsilon_k N}{4} + \frac{\|x_{S^c \cap I^c}^k\|_2^2}{\varepsilon_k}.$$

Hence, using that $\varepsilon_k \leq \sigma_s(x^k)_{\ell_1}/N$, we have shown that

$$\sum_{i \in S^c} \frac{x_i^k}{\max(|x_i^k|, \varepsilon_k)} v_i^k \leq 2\| (x_*)_{S^c} \|_1 - \|v_{S^c}^k\|_1 + \frac{\varepsilon_k N}{4}$$
$$\leq 2\| (x_*)_{S^c} \|_1 - \|v_{S^c}^k\|_1 + \frac{\sigma_s(x^k)_{\ell_1}}{4}$$
$$\leq 2\| (x_*)_{S^c} \|_1 - \|v_{S^c}^k\|_1 + \frac{\|x_{S^c}^k\|_1}{4}$$
$$\leq 2\| (x_*)_{S^c} \|_1 - \|v_{S^c}^k\|_1 + \frac{\|v_{S^c}^k\|_1}{4} + \frac{\|(x_*)_{S^c}\|_1}{4}$$
$$= \frac{9}{4}\| (x_*)_{S^c} \|_1 - \frac{3}{4}\|v_{S^c}^k\|_1,$$

where we used the triangular inequality for the vector $v^k = x^k - x_*$ on the set $S^c$ and the fact that $\sigma_s(x^k)_{\ell_1} \leq \|x_{S^c}^k\|_1$. Hence, by adding up terms we obtain that

$$\langle \nabla \mathcal{J}_{\varepsilon_k}(x^k), v^k \rangle \leq \frac{9}{4}\| (x_*)_{S^c} \|_1 - \left(\frac{3}{4} - \rho_s\right) \|v_{S^c}^k\|_1 \leq -(\beta - \rho_s)\|v_{S^c}^k\|_1.$$

Here, we have set $\beta = 3/4$ in the case that $\sigma_s(x_*)_{\ell_1} = 0$ and $\beta = 1/4$ else. Moreover, we used the assumption $\|(x_*)_{S^c}\|_1 \leq \frac{2}{9}\|v_{S^c}^k\|_1$.

**Part II: Bounding the quadratic term** $\langle v^k, \text{diag}(w_{\varepsilon_k}(x^k))v^k \rangle$

In order bound the quadratic term in (19) we first decompose it into two parts

$$\langle v^k, \text{diag}(w_{\varepsilon_k}(x^k))v^k \rangle = \sum_{i=1}^N \frac{(v_i^k)^2}{\max(|x_i^k|, \varepsilon_k)} = \sum_{i \in S} \frac{(v_i^k)^2}{\max(|x_i^k|, \varepsilon_k)} + \sum_{i \in S^c} \frac{(v_i^k)^2}{\max(|x_i^k|, \varepsilon_k)}. \quad (20)$$

For the first summand, we note that

$$\sum_{i \in S} \frac{(v_i^k)^2}{\max(|x_i^k|, \varepsilon_k)} \leq \frac{\|v_S^k\|_1 \|v_S^k\|_\infty}{\varepsilon_k} \leq \rho_s \frac{\|v_{S^c}^k\|_1 \|v^k\|_\infty}{\varepsilon_k} \leq \frac{\|v_{S^c}^k\|_1 \|v^k\|_\infty}{\varepsilon_k}. \quad (21)$$

For the second summand, it holds that

$$\sum_{i \in S^c} \frac{\left(v_i^k\right)^2}{\max(|x_i^k|, \varepsilon_k)} \leq \frac{\|v_{S^c}^k\|_\infty \|v_{S^c}^k\|_1}{\varepsilon_k} \leq \frac{\|v_{S^c}^k\|_1 \|v^k\|_\infty}{\varepsilon_k}, \quad (22)$$

Hence, by adding (21) and (22) up, it follows that

$$\langle v^k, \text{diag}(w_{\varepsilon_k}(x^k))v^k \rangle \leq 2\frac{\|v_{S^c}^k\|_1 \|v^k\|_\infty}{\varepsilon_k}.$$

Next, we note that

$$\|v^k\|_\infty \leq \rho_1 \|v^k\|_1 \leq \rho_1 (1 + \rho_s) \|v_{S^c}^k\|_1 \leq 2\rho_1 \|v_{S^c}^k\|_1.$$

Hence, we have shown that

$$\langle v^k, \text{diag}(w_{\varepsilon_k}(x^k))v^k \rangle \leq 4\rho_1 \frac{\|v_{S^c}^k\|_1^2}{\varepsilon_k}.$$

**Part III: Combining the bounds to obtain decrease in $k$-th step:**

Inserting the bounds of Part I and Part II into (19) we obtain

$$Q_{\varepsilon_k}(x^k + tv^k, x^k) - \mathcal{J}_{\varepsilon_k}(x^k) \leq -tb + t^2 a =: h(t), \quad (23)$$

where the function $h : \mathbb{R} \to \mathbb{R}$ is a quadratic polynomial with coefficients $b = (\beta - \rho_s) \|v_{S^c}^k\|_1$ and $a = 4\rho_1 \frac{\|v_{S^c}^k\|_1^2}{\varepsilon_k}$. We observe that the minimizer of $h$ is given by $t = \frac{b}{2a}$. Inserting this into $h$, we obtain that

$$h\left(\frac{b}{2a}\right) = -\frac{b^2}{4a} = -\frac{(\beta - \rho_s)^2 \|v_{S^c}^k\|_1^2 \varepsilon_k}{16\rho_1 \|v_{S^c}^k\|_1^2} = -\frac{(\beta - \rho_s)^2}{16\rho_1} \varepsilon_k. \quad (24)$$

Combining this with (6), we obtain, for $t = \frac{b}{2a}$,

$$\mathcal{J}_{\varepsilon_{k+1}}(x^{k+1}) - \mathcal{J}_{\varepsilon_k}(x^k) \leq Q_{\varepsilon_k}(x^{k+1}, x^k) - \mathcal{J}_{\varepsilon_k}(x^k)$$

$$\leq Q_{\varepsilon_k}(x^k + tv^k, x^k) - \mathcal{J}_{\varepsilon_k}(x^k) \leq -\frac{(\beta - \rho_s)^2}{16\rho_1} \varepsilon_k.$$

Hence, by rearranging terms it follows that

$$\mathcal{J}_{\varepsilon_{k+1}}(x^{k+1}) - \|x_*\|_1 \leq \mathcal{J}_{\varepsilon_k}(x^k) - \|x_*\|_1 - \frac{(\beta - \rho_s)^2}{16\rho_1} \varepsilon_k. \quad (25)$$

In order to proceed, we need to bound $\varepsilon_k$ from below. For that, we note that

$$\varepsilon_k = \min\left(\varepsilon_{k-1}, \frac{\sigma_s(x^k)_{\ell_1}}{N}\right) = \frac{\sigma_s\left(x^\ell\right)_{\ell_1}}{N}$$

for some $\ell \leq k$. By Lemma B.1, we have the following inequality chain

$$N\varepsilon_k = \sigma_s\left(x^\ell\right)_{\ell_1} \geq \frac{1}{3}\left(\mathcal{J}_{\varepsilon^\ell}\left(x^\ell\right) - \|x_*\|_1\right) \geq \frac{1}{3}\left(\mathcal{J}_{\varepsilon^k}\left(x^k\right) - \|x_*\|_1\right),$$

where in the second inequality we have used that, by induction, $\mathcal{J}_{\varepsilon_k}\left(x^k\right) \leq \mathcal{J}_{\varepsilon^\ell}\left(x^\ell\right)$. Plugging this into (25) leads to

$$\mathcal{J}_{\varepsilon_{k+1}}(x^{k+1}) - \|x_*\|_1 \leq \left(1 - \frac{(\beta - \rho_s)^2}{48\rho_1 N}\right)\left(\mathcal{J}_{\varepsilon_k}(x^k) - \|x_*\|_1\right).$$

This finishes the induction step and concludes the proof of Proposition B.3.

$\square$

From Proposition B.3 we can deduce Theorem 3.2, the first main result of this manuscript.

*Proof of Theorem 3.2 .* Recall that by Proposition B.3 we have for all $k \in \mathbb{N}$ that

$$\mathcal{J}_{\varepsilon_k}(x^k) - \|x_*\|_1 \leq \left(1 - \frac{c_{\rho_s}}{\rho_1 N}\right)^k \left(\mathcal{J}_{\varepsilon_0}(x^0) - \|x_*\|_1\right)$$

with a constant $c_{\rho_s} = \frac{(3/4 - \rho_s)^2}{48}$ and where $S$ denotes the set, which contains the $s$ largest entries of $x_*$ in absolute value. By our assumption $\rho_s < 1/2$ it follows that $c_{\rho_s} \geq 1/768$, which implies that inequality (9) holds.

By Lemma B.1 we have that

$$\mathcal{J}_{\varepsilon_0}(x^0) - \|x_*\|_1 \leq 3\sigma_s\left(x^0\right)_{\ell_1} \leq 3\|x^0 - x_*\|_1.$$

Next, we note that, again by Lemma B.1, it holds that

$$\frac{1 - \rho_s}{1 + \rho_s}\|x - x_*\|_1 \leq \mathcal{J}_{\varepsilon_k}(x^k) - \|x_*\|_1.$$

Combining the three inequalities in this proof together with the assumption $\rho_s < 1/2$ yields inequality (10), which finishes the proof. $\square$

Next, we are going to prove the second main result in this manuscript, Theorem A.1, which deals with the approximately sparse case.

*Proof of Theorem A.1.* Recall that

$$\hat{k} := \min\left\{k \in \mathbb{N} : \ \sigma_s\left(x_*\right)_{\ell_1} > \frac{2}{9}\|\left(x_*\right)_{S^c} - x_{S^c}^k\|_1\right\}.$$

Moreover, we note that by Proposition B.3 we have for $k \leq \hat{k}$

$$\mathcal{J}_{\varepsilon_k}(x^k) - \|x_*\|_1 \leq \left(1 - \frac{c_{\rho_s}}{\rho_1 N}\right)^k \left(\mathcal{J}_{\varepsilon_0}(x^0) - \|x_*\|_1\right) \tag{26}$$

with a constant $c_{\rho_s} = \frac{(1/4 - \rho_s)^2}{48}$. Hence, by our assumption $\rho_s < 1/8$ we obtain $c_{\rho_s} \geq 1/3072$ and inequality (12) follows, which proves the first statement. In order to prove the second statement, let $\tilde{k}$

and $k$ be natural numbers, such that $\tilde{k} \leq \hat{k}$ and $k \geq \tilde{k}$ holds. Then we obtain that

$$\frac{1 - \rho_s}{1 + \rho_s} \|x^k - x_*\|_1 - 2\sigma_s (x_*)_{\ell_1} \leq \mathcal{J}_{\varepsilon_k}(x^k) - \|x_*\|_1$$

$$\leq \mathcal{J}_{\varepsilon_{\tilde{k}}}(x^{\tilde{k}}) - \|x_*\|_1$$

$$\leq \left(1 - \frac{c_{\rho_s}}{\rho_1 N}\right)^{\tilde{k}} \left(\mathcal{J}_{\varepsilon_0}(x^0) - \|x_*\|_1\right)$$

$$\leq 3 \left(1 - \frac{c_{\rho_s}}{\rho_1 N}\right)^{\tilde{k}} \sigma_s \left(x^0\right)_{\ell_1},$$

where in the first inequality we applied Lemma B.1. In the second inequality we used that the sequence $\left\{\mathcal{J}_{\varepsilon^\ell}\left(x^\ell\right)\right\}_\ell$ is monotonically decreasing and in the third inequality we used inequality (26). In the fourth inequality we again used Lemma B.1. By rearranging terms and using the assumption $\rho_s < 1/8$ it follows for all integers $\tilde{k}$ and $k$ such that $\tilde{k} \leq \hat{k}$ and $k \geq \tilde{k}$

$$\|x^k - x_*\|_1 \leq 6 \left(1 - \frac{c_{\rho_s}}{\rho_1 N}\right)^{\tilde{k}} \sigma_s \left(x^0\right)_{\ell_1} + 4\sigma_s (x_*)_{\ell_1}. \tag{27}$$

In order to proceed, recall that $S$ denotes the support of the $s$ largest entries of $x_*$. Then we note that

$$\sigma_s \left(x^0\right)_{\ell_1} \leq \|x^0_{S^c}\|_1 \leq \| \left(x^0 - x_*\right)_{S^c} \|_1 + \| (x_*)_{S^c} \|_1 \leq \|x^0 - x_*\|_1 + \sigma_s (x_*)_{\ell_1}. \tag{28}$$

Hence, we have shown that for all integers $\tilde{k}$ and $k$ such that $\tilde{k} \leq \hat{k}$ and $k \geq \tilde{k}$ it holds that

$$\|x^k - x_*\|_1 \leq 6 \left(1 - \frac{c_{\rho_s}}{\rho_1 N}\right)^{\tilde{k}} \|x^0 - x_*\|_1 + 10\sigma_s (x_*)_{\ell_1}. \tag{29}$$

By setting $k = \tilde{k}$, we observe that this implies inequality (13), which shows the second statement. In order to prove the third statement, we will distinguish two cases. For the first case, assume that $\hat{k} \geq \left\lceil \frac{\rho_1 N}{c_{\rho_s}} \log \left(\frac{\|x^0 - x_*\|_1}{\sigma_s(x_*)_{\ell_1}}\right) \right\rceil$. Then for $k \geq \tilde{k} := \left\lceil \frac{\rho_1 N}{c_{\rho_s}} \log \left(\frac{\|x^0 - x_*\|_1}{\sigma_s(x_*)_{\ell_1}}\right) \right\rceil$ it follows from inequality (29) that

$$\|x^k - x_*\|_1 \leq 6 \left(1 - \frac{c_{\rho_s}}{\rho_1 N}\right)^{\frac{\rho_1 N}{c_{\rho_s}} \log \left(\frac{\|x^0 - x_*\|_1}{\sigma_s(x_*)_{\ell_1}}\right)} \|x^0 - x_*\|_1 + 10\sigma_s (x_*)_{\ell_1} \leq 20\sigma_s (x_*)_{\ell_1}.$$

where in the second inequality we have used the elementary inequality $\log (1 + t) \leq t$ for $t > -1$. This shows the third statement in the first case. To prove the second case, assume that $\hat{k} < \left\lceil \frac{\rho_1 N}{c_{\rho_s}} \log \left(\frac{\|x^0 - x_*\|_1}{\sigma_s(x_*)_{\ell_1}}\right) \right\rceil$. Then we can compute that

$$\frac{1 - \rho_s}{1 + \rho_s} \|x^k - x_*\|_1 - 2\sigma_s (x_*)_{\ell_1} \leq \mathcal{J}_{\varepsilon_k}(x^k) - \|x_*\|_1$$

$$\leq \mathcal{J}_{\varepsilon_{\tilde{k}}}(x^{\hat{k}}) - \|x_*\|_1$$

$$\leq 3\sigma_s \left(x^{\hat{k}}\right)_{\ell_1}$$

$$\leq 3\|x^{\hat{k}} - x_*\|_1 + 3\sigma_s (x_*)_{\ell_1}$$

$$\leq 3 (1 + \rho_s) \| \left(x^{\hat{k}} - x_*\right)_{S^c} \|_1 + 3\sigma_s (x_*)_{\ell_1}$$

$$\leq 20\sigma_s (x_*)_{\ell_1}.$$

In the first and third inequality we have used Lemma B.1. In the second inequality we have used the monotonicity of the sequence $\left\{\mathcal{J}_{\varepsilon_k}\left(x^k\right)\right\}_k$. In the fourth inequality we have argued as in inequality (28) and in the fifth inequality we have used the null space property. In the last inequality we have used that by definition of $\hat{k}$ it holds that $\sigma_s (x_*)_{\ell_1} > \frac{2}{9}\| (x_*)_{S^c} - x^{\hat{k}}_{S^c}\|_1$. This shows that also in the second case the third statement holds, which finishes the proof. $\square$

## C Computationally Efficient Weighted Least Squares Updates

As pointed out in Section 2, the weighted least squares step (3) of Algorithm 1 can be computed using the explicit formula provided by the weighted Moore-Penrose inverse such that $x^{k+1} = W_k^{-1} A^* (AW_k^{-1} A^*)^{-1}(y)$ where $W_k = \text{diag}(w_k)$, see e.g. [32, Chapter 15.3]. This leads to the main computational step of solving the positive linear system

$$\left(AW_k^{-1} A^*\right) z = y \tag{30}$$

of size $(m \times m)$, which is solved in practice often by an iterative solver such as the conjugate gradient method [38, 49], especially if the measurement matrix $A$ allows for fast matrix-vector multiplications [30, 67]. The authors of [30] provide conditions on the accuracy of conjugate gradient (CG) solvers of the successive linear systems for an IRLS algorithm for basis pursuit using weights that are given by

$$(w_k)_i = 1/\sqrt{|x_i^k|^2 + \varepsilon_k^2} \tag{31}$$

for each $i \in [n]$ (coinciding with the choice of the weights of [3, 23]).

It was observed that combining IRLS with CG provides a competitive algorithm for sparse recovery if the required accuracy is chosen appropriately [30], however, with the problem that as the IRLS method approaches a sparse solution, and therefore, the smoothing parameter $\varepsilon_k$ becomes small, the linear system matrix $AW_k^{-1} A^*$ suffers more and more from ill-conditioning [30, Section 5.2]. To the best of our knowledge, this numerical issue has not been resolved for any IRLS method for sparse recovery in the literature so far.

In this section, we provide an implementation of the weighted least squares step (3) that avoids the ill-conditioning issue of previous IRLS methods and we sketch why the condition number of the resulting linear system can indeed be bounded under standard assumptions.

The starting point of this implementation is the observation that for weight choice (5) used in this paper, i.e., for

$$(w_k)_i := \frac{1}{\max\left(|x_i^k|, \varepsilon_k\right)}$$

for each $i \in [N]$, unlike for (31), it is possible to write the inverse weight matrix $W_k^{-1} \in \mathbb{R}^{N \times N}$ such that

$$W_k^{-1} = W_{I_k}^{-1} + \varepsilon_k \left(\text{Id}_N - P_{I_k}\right) = \left(W_{I_k}^{-1} - \varepsilon_k P_{I_k}\right) + \varepsilon_k \text{Id}, \tag{32}$$

if $I_k := \{i \in [N] : |x_i^k| > \varepsilon_k\}$ denotes the set $I$ of the proof of Proposition B.3, $W_{I_k}^{-1} \in \mathbb{R}^{N \times N}$ is the diagonal matrix with entries $|x_i^k|$ if $i \in I_k$ and 0 otherwise, and where $P_{I_k}$ denotes the projection matrix such that $P_{I_k} x = x_{I_k}$. In particular, we observe that $(W_k)^{-1}$ is the sum of a scaled identity matrix Id and a (diagonal) matrix with only $s_k := |I_k|$ non-zero entries. Furthermore, due to the update rule (4) of the smoothing parameter $\varepsilon_k$, it can be seen that $s_k$ is small and of the order of $s$ if $\varepsilon_k$ approaches 0, i.e., if the $k$-th iterate $x^k$ of Algorithm 1 has only small coordinates outside a set of $s$ large coordinates.

By the Sherman-Morrison-Woodbury matrix inversion formula [70], cf. Lemma C.1 below, it is therefore possible to reformulate the weighted least squares problem such that $x^{k+1}$ can be computed such that solving a positive definite linear system of size $(s_k \times s_k)$, which furthermore is well-conditioned, becomes the main computational step, avoiding solving the ill-conditioned system (30) in this way. We summarize this implementation in Algorithm 2 below.

The implementation uses the matrix $V \in \mathbb{R}^{m \times N}$ with orthonormal columns which denotes a the projection onto the range space of the measurement matrix $A \in \mathbb{R}^{m \times N}$, as well as the left singular vector matrix $U \in \mathbb{R}^{m \times m}$ of $A$ and the diagonal matrix $\Sigma_A \in \mathbb{R}^{m \times m}$ containing the singular values of $A$. These can be pre-computed before using IRLS, for example via a singular value decomposition of $A$. [6] Likewise, the vector

$$\widetilde{y} = V \Sigma_A^{-1} (U^* y) \tag{33}$$

---

[6] In a large scale setting where $A$ is only accessed through matrix-vector multiplications, providing this information is not necessary as respective information can be replaced by using matrix-vector multiplications with $A$, $(AA^*)^{-1}$ and $A^*$.

can be pre-computed and can be re-used at each outer iteration of IRLS.

In the following, if $I \subset [N]$, we denote by $M_I \in \mathbb{R}^{m \times |I|}$ the restriction of a matrix $M \in \mathbb{R}^{m \times N}$ to the columns indexed by $I$, and by $Q_{I_k} \in \mathbb{R}^{N \times I_k}$ the projector matrix such that $P_{I_k} = Q_{I_k} Q_{I_k}^*$. Furthermore, let $D_{I_k}^{-1} \in \mathbb{R}^{I_k \times I_k}$ be a diagonal matrix such that $(D_{I_k}^{-1})_{ii} = |x_i^k|$ for each $i \in I_k$.

---

**Algorithm 2** Practical implementation of weighted least squares step of IRLS for small $\varepsilon_k$

**Input:** Matrix $V \in \mathbb{R}^{m \times N}$ projecting onto range space of measurement matrix $A$, $\widetilde{y} \in \mathbb{R}^N$ from (33), smoothing parameter $\varepsilon_k$, projection $\gamma_k^{(0)} = Q_{I_k}^* Q_{I_{k-1}}(\gamma_{k-1}) \in \mathbb{R}^{I_k}$ of solution $\gamma_{k-1} \in \mathbb{R}^{I_{k-1}}$ of linear system (34) for previous iteration $k-1$.

1: Compute $h_k^0 = Q_{I_k}^* \widetilde{y} - \left( \varepsilon_k \left( D_{I_k}^{-1} - \varepsilon_k \mathrm{Id} \right)^{-1} + (V^*)_{I_k}^* (V^*)_{I_k} \right) \gamma_k^{(0)} \in \mathbb{R}^{I_k}$.

2: Solve

$$\left( \varepsilon_k \left( D_{I_k}^{-1} - \varepsilon_k \mathrm{Id} \right)^{-1} + (V^*)_{I_k}^* (V^*)_{I_k} \right) \Delta \gamma_k = h_k^0 \tag{34}$$

for $\gamma_k \in \mathbb{R}^{I_k}$ by the *conjugate gradient* method [38, 49].

3: Compute $\gamma_k = \gamma_k^{(0)} + \Delta \gamma_k \in \mathbb{R}^{I_k}$.

4: Compute residual $r_{k+1} := \widetilde{y} - V(V^*)_{I_k}(\gamma_k) \in \mathbb{R}^N$.

5: Set $x^{k+1} = r_{k+1}$.

6: Set $(x^{k+1})_{I_k} = (x^{k+1})_{I_k} + \gamma_k$.

7: **Output:** $x^{k+1} \in \mathbb{R}^N$ and $\gamma_k \in I_k$.

---

Next, we verify that $x^{k+1}$ as computed by Algorithm 2 indeed is a solution of the weighted least squares problem (3).

As a preparation, we state the Sherman-Morrison-Woodbury formula for matrix inversion.

**Lemma C.1** ([70]). *Let $B \in \mathbb{R}^{n \times n}, C \in \mathbb{R}^{k \times k}, E \in \mathbb{R}^{n \times k}$ and $F \in \mathbb{R}^{k \times n}$. Then, it holds that*
$$(ECF^* + B)^{-1} = B^{-1} - B^{-1}E(C^{-1} + F^*B^{-1}E)^{-1}F^*B^{-1}$$

With this, we proceed to the proof of the following statement.

**Lemma C.2.** *If $x^{k+1} \in \mathbb{R}^N$ is the output of Algorithm 2, then $x^{k+1}$ coincides with the solution of the weighted least squares problem* (3).

*Proof of Lemma C.2.* We recall from above that if $x_*^{k+1} \in \mathbb{R}^N$ is the solution of (3), it holds that $x_*^{k+1} = W_k^{-1} A^*(z)$ where $z$ is as in (30). Using (32), we see that
$$AW_k^{-1} A^* = A \left( \left( W_{I_k}^{-1} - \varepsilon_k P_{I_k} \right) + \varepsilon_k \mathrm{Id} \right) A^* = A_{I_k} \left( D_{I_k}^{-1} - \varepsilon_k \mathrm{Id} \right) A_{I_k}^* + \varepsilon_k AA^*.$$

By identifying $B := \varepsilon_k AA^*$, $C := \left( D_{I_k}^{-1} - \varepsilon_k \mathrm{Id} \right)$ and $E = F := A_{I_k}$, and by noting that the matrix $C$ is invertible, since, on the set $I_k$, we have $|x_i^{k+1}| > \varepsilon_k$, we obtain by using Lemma C.1 that
$$\left( A(W_k)^{-1} A^* \right)^{-1} = \varepsilon_k^{-1} Z - \varepsilon_k^{-1} Z A_{I_k} G^{-1} A_{I_k}^* Z,$$
using the notation $Z := (AA^*)^{-1}$ and $G := \varepsilon_k C^{-1} + A_{I_k}^* Z A_{I_k}$.

Therefore, denoting the matrix $\varepsilon_k C^{-1} + A_{I_k}^* Z A_{I_k}$ by $G$, we have

$$z = \left( A(W_k)^{-1} A^* \right)^{-1} y = \varepsilon_k^{-1} Z(y) - \varepsilon_k^{-1} Z A_{I_k} G^{-1} A_{I_k}^* Z(y) = \varepsilon_k^{-1} Z(y - A_{I_k} G^{-1} A_{I_k}^* Z(y)) \tag{35}$$

and therefore

$$
\begin{aligned}
A_{I_k}^*(z) &= \varepsilon_k^{-1} (A_{I_k}^* Z(y) - A_{I_k}^* Z A_{I_k} G^{-1} A_{I_k}^* Z(y)) \\
&= \varepsilon_k^{-1} (A_{I_k}^* Z(y) - A_{I_k}^* Z A_{I_k} \left( \varepsilon_k C^{-1} + A_{I_k}^* Z A_{I_k} \right)^{-1} A_{I_k}^* Z(y)) \\
&= \varepsilon_k^{-1} \left( A_{I_k}^* Z(y) - \left( A_{I_k}^* Z A_{I_k} \pm \varepsilon_k C^{-1} \right) \left( \varepsilon_k C^{-1} + A_{I_k}^* Z A_{I_k} \right)^{-1} A_{I_k}^* Z(y) \right) \\
&= C^{-1} \left( \varepsilon_k C^{-1} + A_{I_k}^* Z A_{I_k} \right)^{-1} A_{I_k}^* Z(y).
\end{aligned}
\tag{36}
$$

Thus, for the $(k+1)$-th iteration of IRLS, we obtain for the solution $x_*^{k+1}$ of (3) the representation

$$
\begin{aligned}
x_*^{k+1} &= W_k^{-1} A^*(z) = \left( \varepsilon_k \operatorname{Id} + Q_{I_k} \left( D_{I_k}^{-1} - \varepsilon_k \operatorname{Id} \right) Q_{I_k}^* \right) A^*(z) \\
&= \left[ \varepsilon_k \operatorname{Id} + Q_{I_k} C Q_{I_k}^* \right] A^* z = \varepsilon_k A^* z + Q_{I_k} C A_{I_k}^*(z) \\
&= \varepsilon_k A^* z + Q_{I_k} C C^{-1} \left( \varepsilon_k C^{-1} + A_{I_k}^* Z A_{I_k} \right)^{-1} A_{I_k}^* Z(y) \\
&= \varepsilon_k A^* z + Q_{I_k} G^{-1} A_{I_k}^* Z(y)
\end{aligned}
\tag{37}
$$

using (36) in the third equality.

Next, if $V$ is as in the input of Algorithm 2, since

$$
A_{I_k}^* Z A_{I_k} = Q_{I_k}^* A^* (AA^*)^{-1} A Q_{I_k} = Q_{I_k}^* V V^* Q_{I_k} = (V^*)_{I_k}^* (V^*)_{I_k},
$$

we observe that the matrix $G$ from above actually coincides with the linear system matrix of (34). Furthermore, if $\gamma_k$ is as in step 3 of Algorithm 2, it satisfies

$$
\gamma_k = \gamma_k^{(0)} + \Delta \gamma_k = \gamma_k^{(0)} + G^{-1} \mathbf{h}_k^0 = \gamma_k^{(0)} + G^{-1} \left( Q_{I_k}^* \widetilde{y} - G \gamma_k^{(0)} \right) = G^{-1} Q_{I_k}^* \widetilde{y},
$$

where we also observe that

$$
Q_{I_k}^* \widetilde{y} = Q_{I_k}^* V \Sigma_A^{-1} U^*(y) = Q_{I_k}^* A^* (AA^*)^{-1} (y) = A_{I_k}^* Z(y).
$$

Using the latter equation, we can identify $z$ of (35) such that

$$
z = \varepsilon_k^{-1} Z(y - A_{I_k} G^{-1} A_{I_k}^* Z(y)) = \varepsilon_k^{-1} Z \left( y - A_{I_k} G^{-1} Q_{I_k}^* \widetilde{y} \right) = \varepsilon_k^{-1} Z \left( y - A_{I_k} \gamma_k \right).
$$

Inserting this into (37), we obtain

$$
\begin{aligned}
x_*^{k+1} &= \varepsilon_k A^* z + Q_{I_k} G^{-1} A_{I_k}^* Z(y) = \varepsilon_k A^* z + Q_{I_k} G^{-1} Q_{I_k}^* \widetilde{y} = \varepsilon_k A^* z + Q_{I_k} \gamma_k \\
&= A^* Z \left( y - A_{I_k} \gamma_k \right) + Q_{I_k} \gamma_k \\
&= V \Sigma_A^{-1} U^* \left( y - U \Sigma_A V^* Q_{I_k} \gamma_k \right) + Q_{I_k} \gamma_k \\
&= \widetilde{y} - V (V^*)_{I_k} \gamma_k + Q_{I_k} \gamma_k \\
&= r_{k+1} + Q_{I_k} \gamma_k,
\end{aligned}
$$

where the residual $r_{k+1}$ is as in step 4 of Algorithm 2.

Comparing the equation $x_*^{k+1} = r_{k+1} + Q_{I_k} \gamma_k$ with the steps 5 and 6 of Algorithm 2, we observe that the output $x^{k+1}$ of Algorithm 2 coincides with $x_*^{k+1}$, which finishes the proof.

$\square$

It is well-known that if the conjugates gradient method is used as an inexact solver using a limited number of iterations only, a sufficient condition for quantifying its accuracy can be achieved by bounding the condition number of the system matrix (see, e.g., [56, Section 5.1]).

In fact, using few iterations of a CG method to solve the system (34) leads typically to quite accurate solutions, in particular if $\varepsilon_k$ is small. This can be seen by analyzing the condition number of the matrix $G$

$$
G = \varepsilon_k \left( D_{I_k}^{-1} - \varepsilon_k \operatorname{Id}_{I_k} \right)^{-1} + A_{I_k}^* Z A_{I_k} =: M_{1,k} + M_{2,k}
\tag{38}
$$

of eq. (34) in Algorithm 2.

We start by bounding the condition number of $M_{2,k}$. For that, let $x$ such that $\|x\|_2 = 1$. We calculate that

$$
\| x^T A_{I_k}^* (AA^*)^{-1} A_{I_k} x \| \leq \| A_{I_k} x \|^2 \| (AA^*)^{-1} \| \leq (1+\delta)^2 \| (AA^*)^{-1} \|,
$$

where in the last inequality we have assumed that $A_{I_k}$ satisfies the restricted isometry property. This implies that

$$
\| M_{1,k} \| = \| A_k^* (AA^*)^{-1} A_k \| \leq (1+\delta)^2 \sigma_{\min}(A)^{-2}.
\tag{39}
$$

In a similar way we can derive that

$$
\sigma_{\min} \left( A_k^* (AA^*)^{-1} A_k \right) \geq (1-\delta)^2 \| A \|^{-2}.
\tag{40}
$$

We note that $M_{1,k}$ is a diagonal matrix with entries given by $\varepsilon_k/(x_i^k - \varepsilon_k)$. Now, by additionally assuming that the iteration $x_k$ is already close to the ground truth and, without loss of generality, by assuming that $k$ is such that $||x^k - x_*||_\infty \le 1/4||x_*||_\infty$, we have

$$||M_{1,k}||_2 = \varepsilon_k/(x_i^k - \varepsilon_k) \le \varepsilon_k/(\frac{3}{4}\min_i|(x_*)_i| - \varepsilon_k). \tag{41}$$

Note that this term becomes arbitrarily small, when $\varepsilon_k$ becomes arbitrarily small.

Hence, for $\varepsilon_k$ small enough it follows that

$$\kappa(G) = \frac{||G||}{\sigma_{\min}(G)} \overset{(a)}{\le} \frac{||M_{1,k}|| + ||M_{2,k}||}{\sigma_{\min}(M_{2,k}) - ||M_{1,k}||} \overset{(b)}{\le} \frac{2(1+\delta)^2||A||^2}{(1-\delta)^2\sigma_{\min}(A)^2} = \frac{2(1+\delta)^2}{(1-\delta)^2}\kappa(A)^2.$$

Here, in (a) we used Weyl's inequality and (b) holds as soon as $\varepsilon_k$ is small enough due to inequalities (39), (40), and (41). Hence, we have shown that if $A$ is well-conditioned, then also the matrix $G$ will be well-conditioned and the CG-method will yield very accurate solutions.

# D    Proofs of technical lemmas

## D.1    Proof of Lemma 2.1

In this section, we prove that the chosen quadratic upper bound satisfies the majorization-minimization property.

*Proof.* We prove each of the three statements separately.

1. Let $x \in \mathbb{R}^N$. Then the $i$-th coordinate of $\mathrm{diag}(w_\varepsilon(x))x$ is given by

$$(\mathrm{diag}(w_\varepsilon(x))x)_i = \begin{cases} \frac{x_i}{|x_i|} = \mathrm{sgn}(x_i), & \text{if } |x_i| > \varepsilon, \\ \frac{x_i}{\varepsilon}, & \text{if } |x_i| \le \varepsilon. \end{cases} = j'_\varepsilon(x_i) = (\nabla\mathcal{J}_\varepsilon(x))_i,$$

   where $\mathcal{J}_\varepsilon(x)$ is the gradient of $\mathcal{J}_\varepsilon$ at $x$.

2. This follows directly from the definition of $Q_\varepsilon(x,z)$ and by setting $x = z$.

3. We define $I := \{i \in [N] : |x_i| > \varepsilon\}$ and write the difference $Q_\varepsilon(z,x) - \mathcal{J}_\varepsilon(z)$ as

$$Q_\varepsilon(z,x) - \mathcal{J}_\varepsilon(z) = \frac{1}{2}\left(\langle z, \mathrm{diag}(w_\varepsilon(x))z\rangle - \langle x, \mathrm{diag}(w_\varepsilon(x))x\rangle\right)$$

$$= \sum_{i\in I}\left(\frac{1}{2}|x_i| + \frac{1}{2}\frac{z_i^2}{|x_i|} - j_\varepsilon(z_i)\right) + \sum_{i\in I^c}\left(\frac{1}{2}\varepsilon + \frac{1}{2}\frac{z_i^2}{\varepsilon} - j_\varepsilon(z_i)\right)$$

   and show that each summand of the two sums is non-negative. In particular, if $i \in I$, then assume first that $|z_i| > \varepsilon$. Then

$$\frac{1}{2}|x_i| + \frac{1}{2}\frac{z_i^2}{|x_i|} - j_\varepsilon(z_i) = \frac{1}{2}\left(|x_i| + \frac{z_i^2}{|x_i|}\right) - |z_i| \ge |z_i| - |z_i| = 0$$

   due to inequality $a \le \frac{1}{2}(a^2/b + b)$, which holds for any $b > 0$.

   On the other hand, if $|z_i| \le \varepsilon$, then

$$\begin{aligned}
\frac{1}{2}|x_i| + \frac{1}{2}\frac{z_i^2}{|x_i|} - j_\varepsilon(z_i) &= \frac{1}{2}|x_i| + \frac{1}{2}\frac{z_i^2}{|x_i|} - \frac{1}{2}\left(\frac{z_i^2}{\varepsilon} + \varepsilon\right) \\
&= \frac{1}{2}(|x_i| - \varepsilon) + \frac{1}{2}z_i^2\left(\frac{1}{|x_i|} - \frac{1}{\varepsilon}\right) \\
&\ge \frac{1}{2}(|x_i| - \varepsilon) + \frac{1}{2}\varepsilon^2\left(\frac{1}{|x_i|} - \frac{1}{\varepsilon}\right) \\
&= \frac{1}{2}\left(|x_i| + \frac{\varepsilon^2}{|x_i|}\right) - \varepsilon \ge \varepsilon - \varepsilon = 0,
\end{aligned} \tag{42}$$

where we used that $\frac{1}{|x_i|} - \frac{1}{\varepsilon} < 0$ in the first inequality. In the second inequality, we again used $a \leq \frac{1}{2}(a^2/b + b)$ for any $b > 0$. Now let $i \in I^c$. We again consider the two cases, $|z_i| \leq \varepsilon$ and $|z_i| > \varepsilon$. In the first case we have that $\frac{1}{2}\varepsilon + \frac{1}{2}\frac{z_i^2}{\varepsilon} - j_\varepsilon(z_i) = 0$, and in the second case we have that

$$\frac{1}{2}\varepsilon + \frac{1}{2}\frac{z_i^2}{\varepsilon} - j_\varepsilon(z_i) = \frac{1}{2}\varepsilon + \frac{1}{2}\frac{z_i^2}{\varepsilon} - |z_i| \geq |z_i| - |z_i| = 0,$$

which concludes the proof.

$\square$

## D.2 Proof of Proposition 4.1

In this section, we prove Proposition 4.1, namely, if $x^k$, k-th IRLS iteration, is already close enough from the ground truth, then we would observe that the support would have been already identified and, consequently, the hardest part of the sparse recovery problem would have been solved.

*Proof.* Let $j \in S^c$, where $S$ is the support set of $x_*$. Then

$$|x_j^k| \leq \sum_{i \in S^c} |x_i^k| < \min_{i \in S} |(x_*)_i| - \sum_{i \in S} |x_i^k - (x_*)_i|,$$

using the assumption $\sum_{i \in S^c} |(x^k)_i| + \sum_{i \in S} |x_i^k - (x_*)_i| = \|x^k - x_*\|_1 < \min_{i \in S} |(x_*)_i|$.

On the other hand, for $j \in S$, we can estimate that

$$|x_j^k| = |x_j^k - (x_*)_j + (x_*)_j| \geq |(x_*)_j| - |x_j^k - (x_*)_j| \geq \min_{i \in S} |(x_*)_i| - \sum_{i \in S} |x_i^k - (x_*)_i|.$$

Taking the previous two inequalities together, we conclude that

$$\max_{j \in S^c} |x_j^k| < \min_{j \in S} |x_j^k|,$$

which finishes the proof.

$\square$