# OpenReview forum: "Iteratively Reweighted Least Squares for Basis Pursuit with Global Linear Convergence Rate"
_NeurIPS.cc/2021/Conference — NeurIPS 2021 Spotlight_

### Official Review · Reviewer_DQJN · 2021-07-13

**Rating:** 7
**Confidence:** 4

**Summary:**

The contribution of this paper is to prove a global convergence rate under a null space property -- previous results only established global convergence without a rate, and  a local linear convergence rate.

**Limitations And Societal Impact:**

yes

**Main Review:**

On the strengths:
- This is indeed a new result.
- The global rate deteriorates with $N$ the dimension of the problem, and the authors demonstrate numerically that there are two phases one where their global rate applies and one where the faster local rate of Daubechies et al apply. Numerically, the authors argue that this dependency on $N$ is unavoidable -- I find this interesting (although this seems to come from your $1/N$ factor in the choice of $\epsilon_k$, could a different choice of $\epsilon_k$ lead to better dependency on $N$?).

On the flip side, as a contribution on its own, this does feel somewhat narrow. The NSP is a strong condition in practice and would not hold for typical problems arising in ML and inverse problems. The paper could be improved by drawing some connections to previous convergence results on proximal methods where this two phase convergence property is also established (e.g. Liang et al, "Activity Identification and Local Linear Convergence of Forward--Backward-type Methods." SIAM Journal on Optimization 27.1 (2017): 408-437). Would the result apply under the weaker non-degeneracy assumptions of these works? Finally, as the authors mention, IRLS is applicable beyond just $\ell_1$ regularisation and is extendible to other regularisers such as subquadratic norms, nuclear norm or $\ell_q$ quasi-norms. My general impression is that the results here could be placed in a more general framework (the proof relies on extension of compressed sensing inequalities which are also established beyond the $\ell_1$ setting) to provide a complete picture on the performance of IRLS.

Edit after author responses: Overall, I think this is a nice addition to the literature, and there are possibly interesting followups that the authors could do for a journal version.

**Time Spent Reviewing:**

2

---

> ### Author Response · Authors · 2021-08-11
> **Reply to Reviewer DQJN**
>
> Thank you for your very comprehensive review of our submission, we appreicate it. A point-by-point response to your comments follows below.
>
>   - _"The global rate deteriorates with $N$ the dimension of the problem, and the authors demonstrate numerically that there are two phases one where their global rate applies and one where the faster local rate of Daubechies et al apply. Numerically, the authors argue that this dependency on $N$ is unavoidable -- I find this interesting \(although this seems to come from your $1/N$ factor in the choice of $\varepsilon_k$, could a different choice of $\varepsilon_k$ lead to better dependency on $N$?"_
>
> This is a very good question. The choice of the smoothing parameter update rule, i.e., how $\varepsilon_k$ is precisely defined, plays a fundamental role in the behavior of the IRLS algorithm. Our proof requires the $1/N$ factor in (4), which can be seen, for example, on line 567 in the chain of inequalities (16). If we had an update rule (4) with a factor substantially larger than $1/N$, the convergence of IRLS to an $\ell_1$-minimizer would be compromised, as well as the linear convergence rate. This can be observed in numerical simulations, which we can provide if you are interested. For a similar reason, the update rule in the paper [22] by Daubechies et al. (which showed the fast local linear convergence) includes a similar $1/N$ factor.
>
>
>   - _"On the flip side, as a contribution on its own, this does feel somewhat narrow. The NSP is a strong condition in practice and would not hold for typical problems arising in ML and inverse problems. The paper could be improved by drawing some connections to previous convergence results on proximal methods where this two phase convergence property is also established (e.g. Liang et al, "Activity Identification and Local Linear Convergence of Forward--Backward-type Methods." SIAM Journal on Optimization 27.1 (2017): 408-437). Would the result apply under the weaker non-degeneracy assumptions of these works?"_
>
> We agree that one may consider the NSP as a strong condition in the sense that it is not clear whether it is fulfilled by the measurement matrix $A$ in practical applications, and it is known to be hard to verify.
>
> However, we would like to point out that in some sense, the NSP as defined in Definition 2.2 is necessary and sufficient for the uniform stable recovery of sparse vectors by Basis Pursuit, see for example [1, Theorem 4.12. and Theorem 4.14.] below. In other words, if such a NSP is not fulfilled, it is not possible to uniformly retrieve approximately sparse solutions via $\ell_1$-minimization/Basis Pursuit. We refer to [2, Theorem 4] for an example where IRLS does not convergence to the $\ell_1$-minimizer (in absence of an NSP). Furthermore, we would like to note that the NSP had also been used in the work [22] of Daubechies et al. that established fast local convergence of IRLS for Basis Pursuit.
>
> From a more general perspective, it is the case that the analyses of many algorithms for sparse regression, such as iterative hard thresholding methods [3] or greedy methods, actually require _stronger_ assumptions than the NSP, such as the _restricted isometry property_ or a _restricted strong convexity/smoothness property_, cf. [1, Chapter 6] and [4, Section 7.5].
>
> Thank you a lot for suggesting to put our result into the context of previous convergence results for methods such as the one descibed in the paper by Liang et al. We think that this will indeed improve the paper, and we provide a more comprehensive disucssion in a final version of the manuscript.
>
> The non-degeneracy condition of Liang et al. specified to an $\ell_1$-regularizer $R$ might indeed be weaker than the NSP. Under the assumption of such a condition, the paper shows that a forward-backward method applied to the Lasso problem exhibits local linear convergence, and that after a finte number of iterations, the region of fast convergence is reached. Proposition 3.6(ii) of Liang et al. provides a bound on this number of iterations, which scales proportionally with the _square_ $||x^*-x^0||^2$ of the $\ell_2$-norm difference between initialization and ground truth (the dependence of the other quantities on the problem parameters is unclear to us). On the other hand, our result Theorem 3.2. for IRLS provides a bound on the number of iterations until the fast linear convergence rate is reached that scales proportionally with $\log(||x^{*}-x^0||_1)$, but also proportionally with the dimension $N$ (see discussion after Theorem 3.2.).
>
>   - _"Finally, as the authors mention, IRLS is applicable beyond just regularisation and is extendible to other regularisers such as subquadratic norms, nuclear norm or  quasi-norms. My general impression is that the results here could be placed in a more general framework (the proof relies on extension of compressed sensing inequalities which are also established beyond the  setting) to provide a complete picture on the performance of IRLS."_
>
> Thanks for pointing this out, a very general framework for convergence rates for IRLS algorithms, which contains results about IRLS for $\ell_q$ and/or nuclear norm minimization, would be indeed very desirable.
>
> To the best of our knowledge, in the case of IRLS optimizing $\ell_q$-quasinorms with $0 < q < 1$, there are no global convergence results available at all, as it is difficult to obtain these due to the non-convexity of the objective. In the case of IRLS for optimizing spectral functions, even less is known, as even for IRLS optimizing a nuclear norm (and thus, a convex objective), no local convergence rate analysis is available. The difficulty lies in the fact that spectral functions are not separable, we refer to the state-of-the-art references [5-7] below, which only contain convergence analyses without a convergence rate.
>
> [1] Simon Foucart and Holger Rauhut, [A Mathematical Introduction to Compressive Sensing](https://link.springer.com/chapter/10.1007/978-0-8176-4948-7_1), Birkhäuser, New York, NY, 2013.
>
> [2] Damian Straszak and Nisheeth K. Vishnoi, [Iteratively reweighted least squares and slime mold dynamics: connection and convergence](https://link.springer.com/content/pdf/10.1007/s10107-021-01644-z.pdf), Mathematical Programming (2021): 1-33.
>
> [3] Prateek Jain, Ambuj Tewari, Purushottam Kar, [On Iterative Hard Thresholding Methods for High-dimensional M-Estimation](https://proceedings.neurips.cc/paper/2014/hash/218a0aefd1d1a4be65601cc6ddc1520e-Abstract.html), NeurIPS 2014, 2014.
>
> [4] Prateek Jain and Purushottam Kar, [Non-convex Optimization for Machine Learning](https://arxiv.org/pdf/1712.07897.pdf), Foundations and Trends® in Machine Learning, 10.3-4 (2017): 142-336.
>
> [5] Karthik Mohan, and Maryam Fazel, [Iterative reweighted algorithms for matrix rank minimization](https://www.jmlr.org/papers/volume13/mohan12a/mohan12a.pdf), The Journal of Machine Learning Research, 13.1 (2012): 3441-3473.
>
> [6] Massimo Fornasier, Holger Rauhut, and Rachel Ward, [Low-rank matrix recovery via iteratively reweighted least squares minimization](https://epubs.siam.org/doi/abs/10.1137/100811404), SIAM Journal on Optimization 21.4 (2011): 1614-1640.
>
> [7] Ming-Jun Lai, Yangyang Xu, and Wotao Yin, [Improved iteratively reweighted least squares for unconstrained smoothed \\ell\_q minimization](https://epubs.siam.org/doi/abs/10.1137/110840364), SIAM Journal on Numerical Analysis, 51.2, 927-957 (2013).

---

> > ### Comment · Reviewer_DQJN · 2021-08-25
> > **response to authors**
> >
> > Thanks for the response. Overall, I think this is a nice addition to the literature, and there are possibly interesting followups that the authors could do for a journal version.

---

### Official Review · Reviewer_74ZJ · 2021-07-15

**Rating:** 7
**Confidence:** 3

**Summary:**

The authors propose an iteratively reweighted least squares (IRLS) algorithm for the basis pursuit (BP) problem. The proposed algorithm differs from existing IRLS variants for basis pursuit mainly in terms of the iterative update of the smoothing parameter $\varepsilon_k$. Provided that the measurement matrix $A$ satisfies the null space property (NSP) of order $s$ with constant $\rho_s = \frac12$, the authors prove that their method converges globally at a linear rate and can recover any $s$-sparse vector $x^*$ from an arbitrary initial point $x_0$. The main contribution of the paper is the establishment of a global linear convergence rate for an IRLS algorithm for BP.

**Limitations And Societal Impact:**

Yes.

**Main Review:**

From my point of view, this paper is very clearly and well written. It features a comprehensive introduction to IRLS algorithms and the BP problem, and it provides an appropriate review of related work and theory. To the best of my knowledge, the above-mentioned global convergence result can be considered novel in the sense that for previous IRLS algorithms for BP only local convergence could be established. Moreover, the authors provide an additional theoretical result concerning the locality condition that was previously required to establish local convergence. Namely, they show that this locality condition necessarily requires that the support of the optimal solution has already been identified. I think that this is also an interesting insight.

All in all, I could not identify any serious flaws. Only, in my opinion, it would be desirable that the authors add a comment on the choice of $s$ in step (4) of Algorithm 1.

**Time Spent Reviewing:**

4

---

> ### Author Response · Authors · 2021-08-11
> **Reply to Reviewer 74ZJ**
>
> Thank you a lot for your feedback and your positive assessment of our contribution.
>
> _"Only, in my opinion, it would be desirable that the authors add a comment on the choice of $s$ in step (4) of Algorithm 1."_
>
> That is a great comment, which we will address in the final version of the paper. Indeed, in order to make our theory work, the IRLS algorithm under consideration requires an a priori estimate of the sparsity $s$ of the ground truth of the signal, a piece of information that is also needed by many other methods for sparse reconstruction such as greedy or thresholding methods (cf. [1] below). In practice, an overestimation of $s$ is not a problem for similar numerical results if the overestimation remains within small multiples of the sparsity of the signal.
>
> We note that there are also versions of IRLS which _do not_ require a-priori knowledge of $s$, see for example [2,3] below, as the update rule for the smoothing parameter is chosen differently. It would be interesting to investigate whether it is possible to extend our analysis to IRLS with such a smoothing parameter update.
>
> [1] Simon Foucart and Holger Rauhut, [A Mathematical Introduction to Compressive Sensing](https://link.springer.com/chapter/10.1007/978-0-8176-4948-7_1), Birkhäuser, New York, NY, 2013.
>
> [2] Sergey Voronin and Ingrid Daubechies, [An iteratively reweighted least squares algorithm for sparse regularization](https://www.ams.org/books/conm/693/), in "Functional Analysis, Harmonic Analysis, and Image Processing", p.391-411, American Mathematical Society, 2017, (see also [here](https://arxiv.org/abs/1511.08970)).
>
> [3] Massimo Fornasier, Steffen Peter, Holger Rauhut, and Stephan Worm, [Conjugate gradient acceleration of iteratively re-weighted least squares methods](https://doi.org/10.1007/s10589-016-9839-8), Computational optimization and applications, 65.1 (2016): 205-259.

---

### Official Review · Reviewer_YDtV · 2021-07-16

**Rating:** 7
**Confidence:** 4

**Summary:**

In this paper, the authors study the convergence rate of the Iterative Reweighted Least Square to solve Basis Pursuit. After recalling the existing results and describing the IRLS algorithm with a specific smoothing parameter updates, the authors state their main result showing the global convergence of IRLS under the $\ell_1$-null space property. Finally, some  numerical experiments are presented to corroborate the theoretical results, showing a two stage convergence, one global described by this paper and one local, once the support is identified, which has been studied previously.

**Limitations And Societal Impact:**

See above.

**Main Review:**

Review Summary
---------------

The paper is clear and well written. The results are correct up to my understanding of the proofs. However, I feel that the results are not really practical (**M1**) and the numerical could be improved (**M2**). Overall, I think this is a correct paper but I would recommend publishing it in an optimization venue, as its interest is mainly theoretical. For these reasons, I would recommend rejection.

Major comments and questions
----------------------------


- **M1:** The results are surprising in the sense that sparse recovery is consider to be a slow task in general, with sub-linear convergence rate (e.g. for ISTA in 1/t). However, the rate proposed in theorem.3.2 seems to be very slow. Indeed, for $\rho = 1/100$, with $N=8000$, we get $(1 - \frac{c}{\rho_1 N})^{1000} \approx 0.98$. Note that this is a favorable scenario for $\rho_1$ value and the value reported in the experiment is around $1/38$. Thus, it seems to me that this bounds are not very practical, and it hides the fact that after a finite number of iteration, the local convergence kicks in with a faster rate. This is a common observation that can be done with ISTA, where linear convergence rate have been proven [A] but are not really observed in practice.

- **M2:** The numerical results do not highlight very well the tighness of the analysis. As the results are presented on synthetic datasets, it would be interesting to compute the constant $\rho_1$ and display the theoretical linear rate with the one obtained in practice. I believe this would show that the proposed rate is not very tight but I might be mistaken. Moreover, showing results on problems smaller scale problems where $\rho_s$ can be computed and highlighting how important this assumption is in practice for the convergence of such algorithm would also improve the case of this paper.

Minor comments, nitpicks and typos
----------------------------------

- **m1:** (*l.78*) I think that $\mathbb R^N$ should be $[N]$ here?

- **m2:** (*l.115*) I think what the authors means is that $\epsilon \to \mathcal J_\epsilon(z)$ is monotonously non-decreasing, and $\epsilon_k$ is non-increasing so $k \to \mathcal J_{\epsilon_k}(z)$ is non-increasing with $k$.

- **m3:** (*l.607*) I don't think the assumption "Moreover, assume that $A$ has the $\ell_1$-NSP of order 1 with constant $\rho_1 < 1$." is necessary in Lemma.B.3 as $\rho_1 \le \rho_s$ no?

- **m4:** I think how you obtain the last line in equation block bellow *l.631* could be clarified. It indeed comes from the fact that $Ax^* = Ax$ but this is not obvious form the text. Making the argument explicit somewhere would make the proof easier to read.

- **m5:** In the 2nd equation of the block bellow *l.635*, the $=$ should be a $\le$ I think.

Extra references
----------------

[A] Yen, I. E. H., Hsieh, C. J., Ravikumar, P., & Dhillon, I. S. (2014, March). Constant Nullspace Strong Convexity and Fast Convergence of Proximal Methods under High-Dimensional Settings. In NIPS (pp. 1008-1016).

**Time Spent Reviewing:**

6 hours

---

> ### Author Response · Authors · 2021-08-11
> **Reply to Reviewer YDtV**
>
> We appreciate your thorough assessment and your valuable feedback. A point-by-point response to your comments follows below.
>
> **Major comments:**
>   - _"**M1**: The results are surprising in the sense that sparse recovery is consider to be a slow task in general, with sub-linear convergence rate (e.g. for ISTA in 1/t). However, the rate proposed in theorem.3.2 seems to be very slow. Indeed, for $\rho=1/100$, with $N=8000$, we get $(1 - \frac{c}{\rho_1 N})^{1000} \approx 0.98$. Note that this is a favorable scenario for $\rho_1$ value and the value reported in the experiment is around $1/38$. Thus, it seems to me that this bounds are not very practical."_
>
> The main point of Theorem 3.2. is indeed that the analyzed IRLS method attains a _global_ linear convergence, in contrast to methods with sub-linear convergence such as ISTA, and that it describes a dependence of the linear convergence factor on the dimension $N$ that captures the empirically observed one comparably well.
>
> It is true that our numerical experiments suggest that the constant $c$ in the factor $(1-\frac{c}{\rho_1 N})$ seems to be not very tight, as linear convergence is observed from the very first iterations with a clearly observable factor that is bounded away from $1$ even beyond small choices of $N$. However, we note that constants such as $c$ in $(1-\frac{c}{\rho_1 N})$ are often even not numerically specified in the machine learning literature (see, e.g., [1,2,3] below), as the precise value is typically hard to estimate.
>
> Nevertheless, we note that the value $1/768$ of the constant $c$ in Theorem 3.2. can be replaced by the sharper constant $c_{\rho_s}$ as defined in Proposition B.3. Moreover, by a closer inspection of our proof it is possible to refine this bound even further and to replace $c_{\rho_s}$ by $\frac{\left( \frac{3}{4} -\rho_s \right)^2}{24 \left( 1+\rho_1 \right)} $. For example, by setting $\rho_s=1/5$ and $\rho_1=1/10$ one obtains that $c\ge 1/100 $. (We believe that it should be possible to refine this constant further.)  In the final version of the manuscript, we will add a remark after the main theorem, which will include this sharper constant.
>
> Thank you for this comment, it will allow us to improve the manuscript.
>
>   - _"... and it hides the fact that after a finite number of iteration, the local convergence kicks in with a faster rate."_
>
> In a final version of our paper, we will make it clearer already in Section 3 that a two phase convergence is taking taking place, a global phase with a linear rate of $(1-\frac{c}{\rho_1 N})$ (as described in Theorem 3.2) and a local phase with fast linear convergence, as established in the work [22] of Daubechies et al. and as presented in Proposition 3.1. However, we note that to the best of our knowledge, there is no other result available in the literature quantifying how many iterations of IRLS are need to achieve the basin of fast linear convergence as required in [22] of Daubechies et al. Our manuscript, on the other hand, provides such a bound, see for example the discussion below Theorem 3.2.
>
>   - _"This is a common observation that can be done with ISTA, where linear convergence rate have been proven [A] but are not really observed in practice."_
>
> We would like to point out that in our numerical experiments for IRLS, a global linear rate is indeed observed in practice. For this, we refer to the experiments of Figure 1 and Figure 2a and 2b: The linear convergence factor $\mu(k)$ is below $1$ for each iteration, and even much smaller than $1$ when a standard initialization is used.
>
>   - _"**M2**: The numerical results do not highlight very well the tighness of the analysis. As the results are presented on synthetic datasets, it would be interesting to compute the constant and display the theoretical linear rate with the one obtained in practice. I believe this would show that the proposed rate is not very tight but I might be mistaken. Moreover, showing results on problems smaller scale problems where $\rho_s$ can be computed and highlighting how important this assumption is in practice for the convergence of such algorithm would also improve the case of this paper."_
>
> Thank you for this comment which touches on a question we are also very interested in. As we pointed out in our reply to **M1** above, our analysis is not designed to provide sharp estimates on the constant $c$ in the linear convergence factor $\tilde{\mu} = (1- \frac{c}{\rho_1 N})$. On the other hand, we think that the numerical experiments in Section 4.2. provide sufficient evidence that we essentially capture the dimension dependence on $N$ of the global linear rate, and that this dependence _cannot_ be improved in general. In particular, the experiment that is visualized in Figure 3 indicates that $\tilde{\mu}$ will not scale better than $(1- \frac{c}{N})$ for some constant $c$ (which seems to be around $100$ in the particular setup of the experiment). We are not aware of good algorithms to get accurate estimates for $\rho_1$ (the calculation of which is in general NP-hard to compute [4, Corollary 7]), however, we think that this is not necessary to make our point as we can use the (coarse) bound $\rho_1 \leq 1$ to estimate that $\tilde{\mu} \leq (1- \frac{c}{N})$.
>
> A further exploration of the tightness of our result is definitely of interest and to be explored in future work.
>
>
> **Minor comments:**
>   - _"**m1**: (l.78) I think that $\mathbb{R}^N$ should be $[N]$ here?"_: Indeed. Thank you for pointing out this typo.
>   - _"**m2**: I think what the authors means is that $\varepsilon \mapsto \mathcal{J}$ $\varepsilon(z)$ is monotonously non-decreasing, and $\varepsilon_k$ is non-increasing so $k \mapsto \mathcal{J}$ $\varepsilon_k (z)$ is non-increasing with $k$."_: Indeed. Thank you for pointing out this typo.
>   - _"**m3**: I don't think the assumption "Moreover, assume that $A$ has the $\ell_1$-NSP of order $1$ with constant $\rho_1 < 1$." is necessary in Lemma.B.3 as $\rho_1 \leq \rho_s$ no?"_: Indeed. We will replace this sentence with "Denote by $\rho_1 <1$ the NSP constant of order $1$."
>   - _"**m4**: I think how you obtain the last line in equation block bellow _l.631_ could be clarified. It indeed comes from the fact that $Ax^*=Ax$ but this is not obvious form the text. Making the argument explicit somewhere would make the proof easier to read."_ We agree. We will add a comment, which explains this equality.
>  - _"**m5**: In the 2nd equation of the block bellow _l.635_, the $=$ should be a $\leq$, I think."_ That is correct. Thank you for spotting this inaccuracy.
>
> [1] Yuxin Chen, Emmanuel Candès, [Solving Random Quadratic Systems of Equations Is Nearly as Easy as Solving Linear Systems](https://papers.nips.cc/paper/2015/hash/7380ad8a673226ae47fce7bff88e9c33-Abstract.html), NeurIPS 2015, 739-747.
>
> [2] Qing Qu, Xiao Li, Zhihui Zhu, [A Nonconvex Approach for Exact and Efficient Multichannel Sparse Blind Deconvolution](https://papers.nips.cc/paper/2019/hash/02e656adee09f8394b402d9958389b7d-Abstract.html), NeurIPS 2019, 4015-4026.
>
> [3] Zeyuan Allen-Zhu, Yuanzhi Li, and Zhao Song, [On the Convergence Rate of Training Recurrent Neural Networks](https://papers.nips.cc/paper/2019/hash/0ee8b85a85a49346fdff9665312a5cc4-Abstract.html), NeurIPS 2019, 6676-6688.
>
> [4] Andreas M. Tillmann, Marc E. Pfetsch, [The computational complexity of the restricted isometry property, the nullspace property, and related concepts in compressed sensing](https://doi.org/10.1109/TIT.2013.2290112), IEEE Transactions on Information Theory 60.2 (2013): 1248-1259.
>
> \
> &nbsp;
>
> Please let us know if you have further questions or concerns regarding our submission. We will appreciate it very much if you consider an adjustment of your rating. Thank you!

---

> > ### Comment · Reviewer_YDtV · 2021-08-18
> > **Post rebuttal comment**
> >
> > I thank the authors for their comprehensive feedback.
> >
> > First, I would like to retract the sentence `I would recommend publishing it in an optimization venue`. Re-reading it, I realize it does not carry the message I intended to pass. I think these results are indeed of interest to the NeurIPS community. My main concern when I wrote this was that I feel the manuscript focuses a lot on the theoretical contribution for IRLS and does not really discuss the result in comparison with existing results for concurrent algorithms for basis pursuit, which are to me equally important for a general purpose conference like NeurIPS. Under the condition that the design matrix is full rank on the support, many methods exhibit a linear convergence rate once on the support and I would have like to have a comparison of these different rates and computational complexity.
> > But as mentioned in the rebuttal, most method do not exhibit a *global* linear convergence. I think a discussion of these aspects would make this clearer, and highlight this contribution that I appear to have underestimated in my initial assessment.
> >
> > Then, on the quantification of the bound tightness, I think it is specially important in this context. Indeed, as the support can always be retrieved with a finite (potentially very large) number of iterations, one can always argue that there is a linear convergence rate with a sufficiently large constant (or slow rate). The strength of the proposed analysis is that it correctly capture the behavior before the support is identified and I would have liked to see this numerically. While computing $\rho_1$ is computationally expensive, using a coarse bound to show that the rate is correct would still be possible, as proposed in Figure.3 or with the high probability bound from remark3.3. Adding this bound on figure 1 and 2 would be nice IMO. But re-reading the paper, I feel that Figure.3 is indeed a first step in showing the tightness of the analysis.
> >
> > Therefor, I will raise my rating to accept as I agree that the contribution is interesting and it outweighs the downside of the paper that could be addressed in a follow up paper.

---

### Official Review · Reviewer_pfZ6 · 2021-07-19

**Rating:** 7
**Confidence:** 4

**Summary:**

The paper studies the Basis Pursuit (BP) optimization problem (minimal l1 norm solution to a linear system).
Iterative Reweighted Least Squares (IRLS) is a majorization minimization algorithm to solve this problem.
The paper shows that it converges *globally* and linearly, under a nullspace property assumption (with parameter <1/2).
Existing results only prove local linear convergence rates, first showing that IRLS identifies the support of the solution, after which the rate becomes linear.
The key is a new update rule for the smoothing parameter used in the majorization of |.| by a scaled Huber function.
Experiments highlight the two regimes : global convergence shown in this paper, followed by a linear convergence with a better rate once the iterate enter a basin of attraction of the solution.



**Limitations And Societal Impact:**

.

**Main Review:**

The paper is well-written, the result is new as far as I know. This problem is of interest to the community, given the impact of L1 based methods on ML and signal/image processing in the last 30 years. As highlighted by the authors in their literature review, there has been a wealth of results on the topic of IRLS published at ML venues in the last years.

Comments:
L32 I don't see how BP can be solved with forwrd backward algorithm, as both terms are non smooth. Perhaps the authors meant primal-dual methods (i.e. Chambolle-Pock) ?
Same goes for ADMM, can you please detail the applicability of this method to BP ? I would say both apply to Lasso, but not to BP.

L115 this function is increasing, no ? at least at 0, the higher epsilon, the higher J_espilon(0). Also in 6, LHS, you use increasingness since e_{k+1} \leq e_{k} by 4, right ?

L127 it holds trivially with v=0, why is it needed to exclude this case ?
The formulation L132 is a bit strange to me, I would put with Ax=y *before* "is the unique solution".

L265 could you point to a more precise location in the Foucart and Rauhut book ?

Fig 1 it seems to me that you encounter numerical errors at iteration 64 1e-10 is the 0 roundoff error compared to 1e6 for float64), I would remove the last point for clarity of the graph (which is already a bit long to parse (but very informative)).
However in Fig 2 the "noisy" aspect of last iterations eems to kick in before numerical errors, can you comment on this aspect?

In Fig1 caption why do you say S_k=S is perfect support identification ? As far as I udnerstood it does not mean that x_k restricted to S^c is 0, no?

L279 can you explicitely state m and s for reproducibility ?

In Lemma B.2 you take S as the set of the largest s entries of x^*, this changes in L596 so can you speecify it  L589 ?

L596 it's epsilon not epsilon_k ?



L78 I would say the complement is with respect to \mathbb{N} not R^N ?
L116, if you have one more page, maybe a 1D graph of L1, scaled huber and Q (for k and k+1, with x_k and x_k+1) can help the reader unfamiliar with this classical MM scheme.
L115 use \mapsto not \to
L171:  obtain an multiplicative
L234 epsilon_k not ^k
several time you put an Upper case after a ":" (eg L272, other occurrences)
L322 you have a broken ref
L473 Darbon not Darbons
L587 has the l_1 -NSP holds.
L587 you use z than x in the proof, maybe you can harmonize

**Time Spent Reviewing:**

3

---

> ### Author Response · Authors · 2021-08-11
> **Reply to Reviewer pfZ6**
>
> We appreciate your constructive and very detailed feedback to our submission. A point-by-point response to your comments follows below.
>
>   - _"I don't see how BP can be solved with forward backward algorithm, as both terms are non smooth. Perhaps the authors meant primal-dual methods (i.e. Chambolle-Pock) ? Same goes for ADMM, can you please detail the applicability of this method to BP ? I would say both apply to Lasso, but not to BP."_
>
> Indeed, forward-backward algorithms can be used for Lasso (see, e.g., [1] below) rather than for Basis Pursuit. We will clarify that, and point out the Chambolle-Pock algorithm for basis pursuit. Regarding ADMM for Basis Pursuit, one needs to introduce a dummy variable and write the problem as $f(x)+||z||_1$ subject to $z=x$, where $f(x)$ is the indicator function of the set {$x \in \mathbb{R}^N | Ax=b$}. We refer to page 41 of the book [2] below for more details.
>
>   - _"L115 this function is increasing, no? at least at 0, the higher epsilon, the higher $J_\varepsilon(0)$. Also in 6, LHS, you use increasingness since $\varepsilon$_{k+1} $\leq \varepsilon_{k}$ by 4, right?"_
>
> Thanks for pointing this out. The correct statement is that $\varepsilon \mapsto \mathcal{J}$$\epsilon (z)$ is monotonously **non-decreasing**, and $k \mapsto \varepsilon_k$ is non-increasing so $k \mapsto \mathcal{J}_{\varepsilon_k}(z)$ is **non-increasing** (in $k$). We fixed that.
>
>   - _"L127 it holds trivially with v=0, why is it needed to exclude this case? The formulation L132 is a bit strange to me, I would put with Ax=y before "is the unique solution"."_
>
> In line 127, it is indeed not necessary to exclude the case of $v=0$, as the inequality is fulfilled trivially in this case.
>
> Regarding the formulation in line 132: If we are describing an instance of (P1), strictly speaking, it is necessary to specify both the measurement matrix $A$ and the data vector $y$, which is why we emphasized that ($y$ needs to be chosen as $Ax$) in our wording, as is common in the compressive sensing literature (see, for example, Theorem 4.4 in the book [3] below).
>
>   - _"L265 could you point to a more precise location in the Foucart and Rauhut book?"_
>
> Absolutely. The precise result can be found in Theorem 9.29, Chapter 9, we will add this to the citation.
>
>   - _"Fig 1 it seems to me that you encounter numerical errors at iteration 64 1e-10 is the 0 roundoff error compared to 1e6 for float64), I would remove the last point for clarity of the graph (which is already a bit long to parse (but very informative)). However in Fig 2 the "noisy" aspect of last iterations seems to kick in before numerical errors, can you comment on this aspect?"_
>
> Thanks for pointing out this aspect of Figure 1, we will remove the last iteration in order to avoid confusion. In Figure 2, the ``wiggling'' that can be seen is due to rounding errors and limitations of the floating point calculations, which are probably more pronounced than in Figure 1 due to the larger dimensionality of the vector ($N=16000$ instead of $N=8000$). We would like to point to the definition of $\zeta(k)$, whose value is between $10^{-6}$ and $10^{-8}$ at the last few iterations of the experiment of Figure 2: In $\zeta(k)$, one _divides_ the $\ell_1$-error by the smallest non-zero coordinate of the ground truth $\min_{i \in S} \|(x_{*})_i|$ and this is of order $10^{-4}$ in the experiment. Therefore, in fact, the $\ell_1$-error is much smaller than what the magnitude of $\zeta(k)$ might suggest, which makes the supposed floating arithmetic errors plausible.
>
>   - _"In Fig1 caption why do you say $S_k=S$ is perfect support identification? As far as I understood it does not mean that $x_k$ restricted to $S^c$ is 0, no?"_
>
> That is true, $S_k=S$ does not imply that $x_k$ restricted to $S^c$ is 0 since $S_k$ is just the set of the s largest coordinates of $x_k$. On the other hand, if we know, a priori, that the ground truth is s-sparse, it is natural to define the current support estimate as $S_k$. If $S_k = S$ remains stable for consecutive iterations, it might be fair to say that IRLS retrieves the support for these iterations and we have perfect support identification, which is precisely the case in Figure 1 if $k \geq 18$.
>
>   - _"L279 can you explicitly state m and s for reproducibility?"_
>
> In this experiment, which is visualized in Figure 1, we chose $s=200$ and $m = \lfloor 2s \log(N/s)\rfloor $, as described in lines 264-266.
>
>   - _"In Lemma B.2 you take S as the set of the largest s entries of $x^*$, this changes in L596 so can you specify it L589 ?"_
>
> You are very right, $S$ needs to be introduced already in line 589.
>
>   - _"L596 it's $\varepsilon$ not $\varepsilon_k$ ?"_ Yes, thank you for pointing out this typo.
>
>   - _"L78 I would say the complement is with respect to $\mathbb{N}$ not $R^N$ ?"_ Indeed, Thank you.
>
>   - _"L116, if you have one more page, maybe a 1D graph of L1, scaled huber and Q (for k and $k+1$, with $x_k$ and $x(k+1)$) can help the reader unfamiliar with this classical MM scheme."_
>
> That is a good idea. We will include it in the final version if the space allow for it.
>
>   - _"L115 use $\mapsto not \to$"_: Fixed. Thank you.
>
>   - _"L171: obtain an multiplicative"_: Fixed. Thank you.
>
>   - _"L234 $\varepsilon_k$ not $\varepsilon^k$ several time you put an Upper case after a ":" (eg L272, other occurrences)"_: We fixed all of them. Thank you.
>
>   - _"L322 you have a broken ref:"_ Fixed. Thank you
>
>   - _"L473 Darbon not Darbons:"_ Fixed. Thank you.
>
>   - _"L587 has the $\ell1$-NSP holds. L587 you use z than x in the proof, maybe you can harmonize."_ Thank you for pointing this out. We will implement this improvement of the presentation.
>
>
> [1] Jingwei Liang, Jalal Fadili, and Gabriel Peyré, [Activity Identification and Local Linear Convergence of Forward--Backward-type Methods](https://doi.org/10.1137/16M106340X), SIAM Journal on Optimization 27.1 (2017): 408-437.
>
> [2] Stephen Boyd, Neal Parikh, Eric Chu, Borja Peleato and Jonathan Eckstein, [Distributed Optimization and Statistical Learning via the Alternating Direction Method of Multipliers](https://web.stanford.edu/~boyd/papers/pdf/admm_distr_stats.pdf), _Foundations and Trends in Machine Learning_, 3\(1\):1–122, 2011.
>
> [3] Simon Foucart and Holger Rauhut, [A Mathematical Introduction to Compressive Sensing](https://link.springer.com/chapter/10.1007/978-0-8176-4948-7_1), Birkhäuser, New York, NY, 2013.

---

> > ### Comment · Reviewer_pfZ6 · 2021-08-18
> > **author's rebuttal**
> >
> > This addresses all of my questions and I maintain my recommendation for acceptance.

---

### Decision · Program_Chairs · 2021-09-27

**Decision:**

Accept (Spotlight)

**Comment:**

All the reviewers stressed that this paper presents strong theoretical and numerical contributions, and it should be accepted. They commented that the strengths of the paper (in terms of quality of the exposition and numerical evaluation) outweigh its weaknesses (in particular the use of the NSP). This being said, I strongly recommend that the author follow the recommendations of the reviewers to improve the quality of the paper.